# Learning Generalizable Representations for Reinforcement Learning via Adaptive Meta-Learner of Behavioral Similarities

**Jianda Chen & Sinno Jialin Pan**
Nanyang Technological University, Singapore
`jianda001@e.ntu.edu.sg, sinnopan@ntu.edu.sg`

## Abstract

How to learn an effective reinforcement learning-based model for control tasks from high-level visual observations is a practical and challenging problem. A key to solving this problem is to learn low-dimensional state representations from observations, from which an effective policy can be learned. In order to boost the learning of state encoding, recent works are focused on capturing behavioral similarities between state representations or applying data augmentation on visual observations. In this paper, we propose a novel meta-learner-based framework for representation learning regarding behavioral similarities for reinforcement learning. Specifically, our framework encodes the high-dimensional observations into two decomposed embeddings regarding reward and dynamics in a Markov Decision Process (MDP). A pair of meta-learners are developed, one of which quantifies the reward similarity and the other quantifies dynamics similarity over the correspondingly decomposed embeddings. The meta-learners are self-learned to update the state embeddings by approximating two disjoint terms in on-policy *bisimulation metric*. To incorporate the reward and dynamics terms, we further develop a strategy to adaptively balance their impacts based on different tasks or environments. We empirically demonstrate that our proposed framework outperforms state-of-the-art baselines on several benchmarks, including conventional DM Control Suite, Distracting DM Control Suite and a self-driving task CARLA.

## 1 Introduction

Designing effective reinforcement learning algorithms for learning to control from high-dimensional visual observations is crucial and has attracted more and more attention (Yarats et al., 2021; Laskin et al., 2020a; Schwarzer et al., 2021; Lesort et al., 2018). To learn a policy efficiently from high-dimensional observations, prior approaches first learn an encoder to map high-dimensional observations, e.g., images, to low-dimensional representations, and subsequently train a policy from low-dimensional representations to actions based on various RL algorithms. Therefore, how to learn low-dimensional representations, which are able to provide semantic abstraction for high-dimensional observations, plays a key role.

Early works on deep reinforcement learning train encoders based on various reconstruction losses (Lange & Riedmiller, 2010; Watter et al., 2015; Wahlström et al., 2015; Higgins et al., 2017), which aim to enforce the learned low-dimensional representations to reconstruct the original high-dimensional observations after decoding. Promising results have been achieved in some application domains, such as playing video games, simulated tasks, etc. However, the policy learned on state representations trained with a reconstruction loss may not generalize well to complex environments, which are even though semantically similar to the source environment. The reason is that a reconstruction loss is computed over all the pixels, which results in all the details of the high-dimensional images tending to be preserved in the low-dimensional representations. However, some of the observed details, such as complex background objects in an image, are task-irrelevant and highly environment-dependent. Encoding such details in representation learning makes the learned representation less effective for a downstream reinforcement learning task of interest and less generalizable to new environments. To address the aforementioned issue, some data augmentation

techniques have been proposed to make the learned representations more robust (Yarats et al., 2021; Lee et al., 2020b; Laskin et al., 2020b). However, these approaches rarely consider the properties of a Markov Decision Process (MDP), such as conditional transition probabilities between states given an action, in the representation learning procedure.

Recent research (Zhang et al., 2021; Agarwal et al., 2021; Castro, 2020; Ferns et al., 2004) has shown that the *bisimulation metric* and its variants are potentially effective to be exploited to learn a more generalizable reinforcement learning agent across semantically similar environments. The bisimulation metric measures the "behavioral similarity" between two states based on two terms: 1) reward difference that considers the difference in immediate task-specific reward signals between two states, and 2) dynamics difference that considers the similarity of the long-term behaviors between two states. A general idea of these approaches is to learn an encoder to map observations to a latent space such that the distance or similarity between two states in the latent space approximates their bisimulation metric. In this way, the learned representations (in the latent space) are task-relevant and invariant to environmental details. However, manually specifying a form of distance, e.g., the $L_1$ norm as used in (Zhang et al., 2021), in the latent space may limit the approximation precision for the bisimulation metric and potentially discard some state information that is useful for policy learning. Moreover, existing approaches rarely explore how to learn an adaptive combination of reward and dynamics differences in the bisimulation metric, which may vary in different tasks or environments.

We propose a novel framework for learning generalizable state representations for RL, named Adaptive Meta-learner of Behavioral Similarities (AMBS). In this framework, we design a network with two encoders that map the high-dimensional observations to two decomposed representations regarding rewards and dynamics. For the purpose of learning behavioral similarity on state representations, we introduce a pair of meta-learners that learn similarities in order to measure the reward and the dynamics similarity between two states over the corresponding decomposed state representations, respectively. The meta-learners are self-learned by approximating the reward difference and the dynamics difference in the bisimulation metric. Then the meta-learners update the state representations according to their behavioral distance to the other state representations. Previous approaches with a hand-crafted form of distance/similarity evaluating state encoding in the $L_1$ space are difficult to minimize the approximation error for the bisimulation metric, which may lead to important side information being discarded, e.g., information regarding to Q-value but not relevant to states distance/similarity. Instead, our learned similarities measure two states representations via a neural architecture, where side information can be preserved for policy learning. Our experiments also showed that a smaller approximation loss for similarity learning can be obtained by using the meta-learners. This demonstrates that the proposed meta-learners can overcome the approximation precision issue introduced by the $L_1$ distance in previous approaches and provide more stable gradients for robust learning of state representations for deep RL. Moreover, we explore the impact between the reward and the dynamics terms in the bisimulation metric. We propose a learning-based adaptive strategy to balance the effect between reward and dynamics in different tasks or environments by introducing a learnable importance parameter, which is jointly learned with the state-action value function. Finally, we use a simple but effective data augmentation strategy to accelerate the RL procedure and learn more robust state representations.

The main contributions of our work are 3-fold: 1) we propose a meta-learner-based framework to learn task-relevant and environment-details-invariant state representations; 2) we propose a network architecture that decomposes each state into two different types of representations for measuring the similarities in terms of reward and dynamics, respectively, and design a learnable adaptive strategy to balance them to estimate the bisimulation metric between states; 3) we verify our proposed framework on extensive experiments and demonstrate new state-of-the-art results on background-distraction DeepMind control suite (Tassa et al., 2018; Zhang et al., 2018; Stone et al., 2021) and other visual-based RL tasks.

## 2 RELATED WORK

**Representation learning for reinforcement learning from pixels**   Various existing deep RL methods have been proposed to address the sample-efficiency and the generalization problems for conventional RL from pixel observation. In end-to-end deep RL, neural networks learn representations implicitly by optimizing some RL objective (Mnih et al., 2015; Espeholt et al., 2018). Watter et al.

(2015) and Wahlström et al. (2015) proposed to learn an encoder with training a dynamics model jointly to produce observation representations. Lee et al. (2020a), Hafner et al. (2019), Hafner et al. (2020) and Zhang et al. (2019) aimed to learn an environment dynamics model with the reconstruction loss to compact pixels to latent representations. Gelada et al. (2019) proposed to learn representations by predicting the dynamics model along with the reward, and analyzed its theoretical connection to the bisimulation metric. Agarwal et al. (2021) proposed to learn representations by state distances based on policy distribution. Kemertas & Aumentado-Armstrong (2021) modified the learning of bisimulation metric with intrinsic rewards and inverse dynamics regularization. Castro et al. (2021) replaced the Wasserstein distance in bisimulation metric with a parametrized metric.

**Data Augmentation in reinforcement learning** Laskin et al. (2020b) and Ye et al. (2020b) explored various data augmentation techniques for deep RL, which are limited to transformations on input images. Cobbe et al. (2019) applied data augmentation to domain transfer. Ye et al. (2020a) studied data augmentation on game environments for zero-shot generalization. RandConv (Lee et al., 2020b) proposed a randomized convolutional neural network to generate randomized observations in order to perform data augmentation. A recent work DrQ (Yarats et al., 2021) performs random crop on image observations and provides regularized formulation for updating Q-function. Raileanu et al. (2021) proposed to use data augmentation for regularizing on-policy actor-critic RL objectives. Pitis et al. (2020) proposed a counterfactual data augmentation technique by swapping observed trajectory pairs.

## 3 PRELIMINARIES

We define an environment as a Markov Decision Process (MDP) described by a tuple $\mathcal{M} = (\mathcal{S}, \mathcal{A}, \mathcal{P}, \mathcal{R}, \gamma)$, where $\mathcal{S}$ is the high-dimensional state space (e.g., images), $\mathcal{A}$ is the action space, $\mathcal{P}(\mathbf{s}'|\mathbf{s}, \mathbf{a})$ is the transition dynamics model that captures the probability of transitioning to next state $\mathbf{s}' \in \mathcal{S}$ given current state $s \in \mathcal{S}$ and action $\mathbf{a} \in \mathcal{A}$, $\mathcal{R}$ is the reward function yielding a reward signal $r = \mathcal{R}(\mathbf{s}, \mathbf{a}) \in \mathbb{R}$, and $\gamma \in [0, 1)$ is the discounting factor. The goal of reinforcement learning is to learn a policy $\pi(\mathbf{a}|\mathbf{s})$ that maximizes the expected cumulative rewards: $\max_\pi \mathbb{E}[\sum_t \gamma^t r(\mathbf{s}_t, \mathbf{a}_t)|a_t \sim \pi(\cdot|\mathbf{s}_t), \mathbf{s}'_t \sim \mathcal{P}(\mathbf{s}'|\mathbf{s}, \mathbf{a})]$. In the scope of this paper, we do not consider partial observability, and use stacked consecutive frames as the fully observed states.

**Soft Actor-Critic (SAC)** is a widely used off-policy model-free reinforcement learning algorithm (Haarnoja et al., 2018). SAC aims to maximize the entropy objective (Ziebart, 2010) which is the reinforcement learning objective augmented with an entropy term $J(\pi) = \sum_t \mathbb{E}[r(\mathbf{s}_t, \mathbf{a}_t) + \alpha \mathcal{H}(\pi(\cdot|\mathbf{s}_t))]$. In order to maximize the objective, SAC learns a state-value function $Q_\theta(\mathbf{s}, \mathbf{a})$, a stochastic policy $\pi_\psi(\mathbf{a}|\mathbf{s})$ and the temperature $\alpha$, where $\theta$ and $\psi$ are the parameters of $Q_\theta$ and $\pi_\psi$, respectively. The Q-function is trained by minimizing the squared soft Bellman residual

$$J_Q(\theta) = \mathbb{E}_{(\mathbf{s}_t, \mathbf{a}_t) \sim \mathcal{D}} \left[ \frac{1}{2} \left( Q_\theta(\mathbf{s}_t, \mathbf{a}_t) - \left( r(\mathbf{s}_t, \mathbf{a}_t) + \gamma \mathbb{E}_{\mathbf{s}_{t+1} \sim \mathcal{P}(\cdot|\mathbf{s}_t, \mathbf{a}_t)}[V_{\bar{\theta}}(\mathbf{s}_{t+1})] \right) \right)^2 \right], \quad (1)$$

where $\mathcal{D}$ is the dataset or replay buffer storing the transitions, and $V_{\bar{\theta}}$ is the value function parameterized by $\bar{\theta}$. The parameters $\psi$ of policy is learned by maximizing the following objective

$$J_\pi(\psi) = \mathbb{E}_{\mathbf{s}_t \sim \mathcal{D}} \left[ \mathbb{E}_{\mathbf{a}_t \sim \pi_\psi}[\alpha \log(\pi_\psi(\mathbf{a}|\mathbf{s})) - Q_\theta(\mathbf{s}_t, \mathbf{a}_t)] \right]. \quad (2)$$

**The Bisimulation Metric** defines a pseudometric $d : \mathcal{S} \times \mathcal{S} \to \mathbb{R}$,[1] where $d$ quantifies the behavioral similarity of two discrete states (Ferns et al., 2004). An extension to both continuous and discrete state spaces has also been developed (Ferns et al., 2011). A variant of the bisimulation metric proposed in Castro (2020) defines a metric w.r.t. a policy $\pi$, which is known as the on-policy bisimulation metric. It removes the requirement of matching actions in the dynamics model but focuses on the policy $\pi$, which is able to better capture behavioral similarity for a specific task. Because of this property, in this paper, we focus on the $\pi$-*bisimulation metric* (Castro, 2020):

$$F^\pi(d)(\mathbf{s}_i, \mathbf{s}_j) = (1 - c)|\mathcal{R}^\pi_{\mathbf{s}_i} - \mathcal{R}^\pi_{\mathbf{s}_j}| + cW_1(d)(\mathcal{P}^\pi_{\mathbf{s}_i}, \mathcal{P}^\pi_{\mathbf{s}_j}). \quad (3)$$

where $\mathcal{R}^\pi_{\mathbf{s}} := \sum_{\mathbf{a}} \pi(\mathbf{a}|\mathbf{s})\mathcal{R}(\mathbf{s}, \mathbf{a})$ and $\mathcal{P}^\pi_{\mathbf{s}} := \sum_{\mathbf{a}} \pi(\mathbf{a}|\mathbf{s})\mathcal{P}(\cdot|\mathbf{s}, \mathbf{a})$. The mapping $F^\pi : \mathfrak{met} \to \mathfrak{met}$, where $\mathfrak{met}$ denotes the set of all pseudometrics in state space $\mathcal{S}$ and $W_1(d)(\cdot, \cdot)$ denotes the 1-Wasserstein distance given the pseudometric $d$. Then $F^\pi$ has a least fixed point denoted by $d_*$. *Deep*

---

[1]If the pseudometric $d$ of two states is 0, then the two states belong to an equivalence relation.

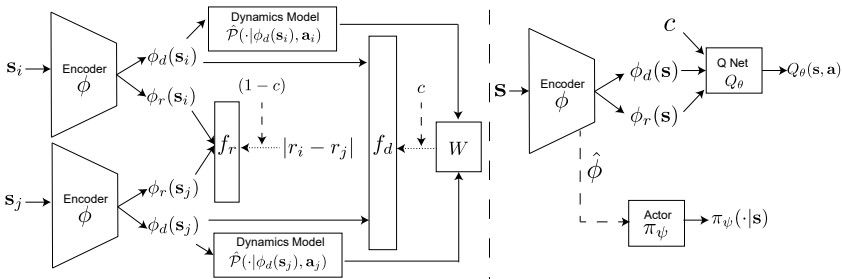

Figure 1: Architecture of our AMBS framework. The dotted arrow represents the regression target and the dash arrow means stop gradient. **Left**: the learning process of meta-learner. **Right**: the model architecture for SAC with adaptive weight $c$ which is jointly learned with SAC objective.

*Bisimulation for Control (DBC)* (Zhang et al., 2021) is to learn latent representations such that the $L_1$ distances in the latent space are equal to the $\pi$-bisimulation metric in the state space:

$$\arg\min_\phi \left( \|\phi(\mathbf{s}_i) - \phi(\mathbf{s}_j)\|_1 - |\mathcal{R}^\pi_{\mathbf{s}_i} - \mathcal{R}^\pi_{\mathbf{s}_j}| - \gamma W_2(d)(\mathcal{P}^\pi_{\mathbf{s}_i}, \mathcal{P}^\pi_{\mathbf{s}_j}) \right)^2,$$

where $\phi$ is the state encoder, and $W_2(d)(\cdot, \cdot)$ denotes the 2-Wasserstein distance. DBC is combined with the reinforcement learning algorithm SAC, where $\phi(\mathbf{s})$ is the input for SAC.

## 4    ADAPTIVE META-LEARNER OF BEHAVIORAL SIMILARITIES

In this section, we propose a framework named Adaptive Meta-learner of Behavioral Similarities (AMBS) to learn generalizable states representation regarding the $\pi$-bisimulation metric. The learning procedure is demonstrated in Figure 1. Observe that the $\pi$-bisimulation metric is composed of two terms: $|\mathcal{R}^\pi_{\mathbf{s}_i} - \mathcal{R}^\pi_{\mathbf{s}_j}|$, which computes the difference of rewards between states, and $W_2(d)(\mathcal{P}^\pi_{\mathbf{s}_i}, \mathcal{P}^\pi_{\mathbf{s}_j})$, which computes the difference of the outputs of dynamics model between states. We propose a network architecture which contains encoders to transform the high-dimensional visual observations to two decomposed encodings regarding rewards and dynamics. We develop two meta-learners, one of which quantifies the reward difference on reward representations and the other captures dynamics distance (Section 4.1). Each meta-learner is self-learned to update the corresponding state representations by learning to approximate a term in the $\pi$-bisimulation metric, respectively. Rather than enforcing the $L_1$ distance between embedded states to be equal to the $\pi$-bisimulation metric, our meta-learners are able to use a more flexible form of similarity, i.e., a well-designed non-linear neural network, with each similarity evaluates the reward difference or dynamics difference between states beyond the original Euclidean space. Moreover, we propose a strategy for learning to combine the outputs of the two learned similarities in a specific environment (Section 4.2). We introduce a learnable weight for the combination and such weight is adaptively learned together with the policy learning procedure (in this work we use SAC as the base reinforcement learning algorithm). In addition, we also use a data augmentation technique to make the learned representations more robust (Section 4.3). The whole learning procedure is trained in an end-to-end manner.

### 4.1    META-LEARNERS FOR DECOMPOSED REPRESENTATIONS

We design a network architecture with two encoders, $\phi_r$ and $\phi_d$, to encode decomposed features from visual observations as shown in the left side of Figure 1. Each encoder maps a high-dimensional observation to a low-dimensional representation, i.e., $\phi_r : \mathcal{S} \to \mathcal{Z}_r$ capturing reward-relevant features and $\phi_d : \mathcal{S} \to \mathcal{Z}_d$ capturing dynamics-relevant features. We design two meta-learners, $f_r$ and $f_d$, where $f_r$ learns to measure reward difference and $f_d$ aims to measure dynamics difference between two state representations. Specifically, each meta-learner takes two embedded states as input and outputs the corresponding similarity. The procedure is summarized as follows.

$$\mathbf{s}_i, \mathbf{s}_j \to \phi_*(\mathbf{s}_i), \phi_*(\mathbf{s}_j) \to f_*(\phi_*(\mathbf{s}_i), \phi_*(\mathbf{s}_j)), \text{ where } * \in \{r, d\}.$$

Note that the aggregation of the outputs of the two encoders $\phi_r$ and $\phi_d$ of each observation (i.e., state)is also fed into SAC to learn a policy (the right side of Figure 1). Each self-learned meta-learner learns to capture similarity by approximating a distance term in the $\pi$-bisimulation metric, respectively, and provides stable gradients for updating $\phi_r$ or $\phi_d$ according to the distance to the other state representations. The details of network architecture can be found in Appendix B.2.

As discussed in Section 1, restricting the form of distance to be the $L_1/L_2$ norm in the latent space may limit the approximation precision. As shown in Figure 2, the $L_1$ and the $L_2$ norm for measuring the distance of two latent representations $\phi(\mathbf{s}_i)$ and $\phi(\mathbf{s}_j)$ lead to large regression losses during RL training, which destabilize the representation learning and consequently decrease final performance. Besides, such hand-crafted distances make semantically similar states encoding close to each other in terms of the $L_1$ norm in Euclidean space, which however may lose part of useful information, e.g., policy-relevant information but not immediately regarding to the distance.

To overcome the aforementioned limitations, we propose to exploit meta-learners $f_r$ and $f_d$ to learn similarities. By learning with a regression target, the similarity learned by $f_*$ is easier to converge to the target and leads to faster descent tendency of regression loss (as shown in Figure 2 where y-axis is log-scale). Such a property provides more stable gradients for state representation comparing to the $L_1/L_2$ norm, and therefore the meta-learner $f_*$ is able to guide the process of updating the state encoder. Besides, $f_*$ is a non-linear transformation that evaluates the state similarity in a more flexible space instead of the original Euclidean space. Consequently, it is able to preserve more task-relevant information in state representation that is required for further SAC policy learning.

As the goal of meta-learners is to approximate different types of similarities in the $\pi$-bisimulation metric, we design the loss functions for the meta-learners using the mean squared error:

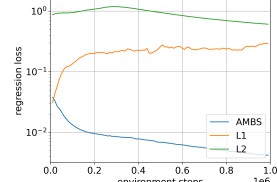

$$\ell(f_r, \phi_r) = \left( f_r(\phi_r(\mathbf{s}_i), \phi_r(\mathbf{s}_j)) - |\mathcal{R}_{\mathbf{s}_i}^\pi - \mathcal{R}_{\mathbf{s}_j}^\pi| \right)^2, \qquad (4)$$

$$\ell(f_d, \phi_d) = \left( f_d(\phi_d(\mathbf{s}_i), \phi_d(\mathbf{s}_j)) - W_2(\mathcal{P}_{\mathbf{s}_i}^\pi, \mathcal{P}_{\mathbf{s}_j}^\pi) \right)^2. \qquad (5)$$

To make the resultant optimization problems compatible with SGD updates with transitions sampled from replay buffer, we replace the reward metric term $|\mathcal{R}_{\mathbf{s}_i}^\pi - \mathcal{R}_{\mathbf{s}_j}^\pi|$ by the distance of rewards in the sampled transitions in (4), and use a learned parametric dynamics model $\hat{\mathcal{P}}$ to approximate the true dynamics model $\mathcal{P}_{\mathbf{s}}^\pi$ in (5). We also use 2-Wasserstein distance $W_2$ in (5) (as in DBC) because it has a closed form for Gaussian distributions. Denote by $(\mathbf{s}_i, \mathbf{a}_i, r_i, \mathbf{s}_i')$ and $(\mathbf{s}_j, \mathbf{a}_j, r_j, \mathbf{s}_j')$ two transitions sampled from the replay buffer. The loss functions (4) and (5) can be revised as follows,

Figure 2: The regression loss among different forms of distances.

$$\ell(f_r, \phi_r) = \left( f_r(\phi_r(\mathbf{s}_i), \phi_r(\mathbf{s}_j)) - |r_i - r_j| \right)^2, \qquad (6)$$

$$\ell(f_d, \phi_d) = \left( f_d(\phi_d(\mathbf{s}_i), \phi_d(\mathbf{s}_j)) - W_2(\hat{\mathcal{P}}(\cdot|\phi_d(\mathbf{s}_i), \mathbf{a}_i), \hat{\mathcal{P}}(\cdot|\phi_d(\mathbf{s}_j), \mathbf{a}_j)) \right)^2, \qquad (7)$$

where $\hat{\mathcal{P}}(\cdot|\phi_d(\mathbf{s}_*), \mathbf{a}_*)$ is a learned probabilistic dynamics model, which is implemented as a neural network that takes the dynamics representation $\phi_d(\mathbf{s}_*)$ and an action $\mathbf{a}_*$ as input, and predicts the distribution of dynamics representation of next state $\mathbf{s}_*'$. The details of $\hat{\mathcal{P}}$ can be found Appendix A.

## 4.2 BALANCING IMPACT OF REWARD AND DYNAMICS

The $\pi$-bisimulation metric (3) is defined as a linear combination of the reward and the dynamics difference. Rather than considering the combination weight as a hyper-parameter, which needs to be tuned in advance, we introduce a learnable parameter $c \in (0, 1)$ to adaptively balance the impact of the reward and the dynamics in different environments and tasks.

As the impact factor $c$ should be automatically determined when learning in a specific task or environment, we integrate the learning of $c$ into the update of the Q-function in SAC such that the value of $c$ is learned from the state-action value which is highly relevant to a specific task and environment. To be specific, we concatenate the low-dimensional representations $\phi_r(\mathbf{s})$ and $\phi_d(\mathbf{s})$ weighted by $1 - c$ and $c$, where $1 - c$ and $c$ are output of a softmax to ensure $c \in (0, 1)$. The loss for the Q-function is revised as follows, which takes $\phi_r(\mathbf{s})$, $\phi_d(\mathbf{s})$, weight $c$ and action $a$ as input:

$$J_Q(\theta, c, \phi_r, \phi_d) = \mathbb{E}_{(\mathbf{s}, \mathbf{a}, \mathbf{s}') \sim \mathcal{D}} \left[ \frac{1}{2} \left( Q_\theta(\phi_r(\mathbf{s}), \phi_d(\mathbf{s}), c, \mathbf{a}) - (r(\mathbf{s}, \mathbf{a}) + \gamma V(\mathbf{s}')) \right)^2 \right], \qquad (8)$$

where $Q_\theta$ is the Q-function parameterized by $\theta$ and $V(\cdot)$ is the target value function.

Moreover, to balance the learning of the two meta-learners $f_r$ and $f_d$, we jointly minimize the regression losses of $f_r$ and $f_d$ ((6) and (7), respectively) with the balance factor $c$. Thus, the loss

function is modified as follows,

$$\ell(\Theta) = (1 - c)\ell(f_r, \phi_r) + c\ell(f_d, \phi_d), \text{ where } \Theta = \{f_r, f_d, \phi_r, \phi_d\}. \tag{9}$$

Note that $c$ is already learned along with the Q-function, we stop gradient of $c$ when minimizing the loss (9). This is because if we perform gradient descent w.r.t. $c$, it may only capture which loss ($\ell(f_r, \phi_r)$ or $\ell(f_d, \phi_d)$) is larger/smaller and fail to learn the quantities of their importance. We also stop gradient for the dynamics model $\hat{\mathcal{P}}$ since $\hat{\mathcal{P}}$ only predicts one-step dynamics.

## 4.3 Overall Objective with Data Augmentation

In this section, we propose to integrate a data augmentation strategy into our proposed framework to learn more robust state representations. We follow the notations of an existing work on data augmentation for reinforcement learning, DrQ (Yarats et al., 2021), and define a state transformation $h : \mathcal{S} \times \mathcal{T} \to \mathcal{S}$ that maps a state to a data-augmented state, where $\mathcal{T}$ is the space of parameters of $h$. In practice, we use a random crop as the transformation $h$ in this work. Then $\mathcal{T}$ contains all possible crop positions and $\mathbf{v} \in \mathcal{T}$ is one crop position drawn from $\mathcal{T}$. We apply DrQ to regularize the objective of the Q-function (see Appendix for the full objective of Q-function).

Besides, by using the augmentation transformation $h$, we rewrite the loss of representation learning and similarity approximation in (9) as follows,

$$
\begin{aligned}
\ell(\Theta) =& (1 - c)\left(f_r\left(\phi_r(\mathbf{s}_i^{(1)}), \phi_r(\mathbf{s}_j^{(1)})\right) - |r_i - r_j|\right)^2 \\
&+ c\left(f_d\left(\phi_d(\mathbf{s}_i^{(1)}), \phi_d(\mathbf{s}_j^{(1)})\right) - W_2\left(\hat{\mathcal{P}}(\cdot|\phi_d(\mathbf{s}_i^{(1)}), \mathbf{a}_i), \hat{\mathcal{P}}(\cdot|\phi_d(\mathbf{s}_j^{(1)}), \mathbf{a}_j)\right)\right)^2 \\
&+ (1 - c)\left(f_r\left(\phi_r(\mathbf{s}_j^{(2)}), \phi_r(\mathbf{s}_i^{(2)})\right) - |r_i - r_j|\right)^2 \\
&+ c\left(f_d\left(\phi_d(\mathbf{s}_j^{(2)}), \phi_d(\mathbf{s}_i^{(2)})\right) - W_2\left(\hat{\mathcal{P}}(\cdot|\phi_d(\mathbf{s}_i^{(1)}), \mathbf{a}_i), \hat{\mathcal{P}}(\cdot|\phi_d(\mathbf{s}_j^{(1)}), \mathbf{a}_j)\right)\right)^2,
\end{aligned}
\tag{10}
$$

where $\mathbf{s}_*^{(k)} = h(\mathbf{s}_*, \mathbf{v}_*^{(k)})$ is the transformed state with parameters $\mathbf{v}_*^{(k)}$. Specifically, $\mathbf{s}_i^{(1)}$, $\mathbf{s}_j^{(1)}$, $\mathbf{s}_i^{(2)}$ and $\mathbf{s}_j^{(2)}$ are transformed states with parameters of $\mathbf{v}_i^{(1)}$, $\mathbf{v}_j^{(1)}$, $\mathbf{v}_i^{(2)}$ and $\mathbf{v}_j^{(2)}$ respectively, which are all drawn from $\mathcal{T}$ independently. The first two terms in (10) are similar to the two terms in (9), while the observation $\mathbf{s}_*$ in (9) is transformed to $\mathbf{s}_*^{(1)}$. For the last two terms in (10), the observation is transformed by

---

**Algorithm 1** AMBS+SAC

1: **Input:** Replay Buffer $\mathcal{D}$, initialized $\Theta = \{f_r, \phi_r, f_d, \phi_d\}$, Q network $Q_\theta$, actor $pi_\psi$, target Q network $Q_{\bar{\theta}}$,
2: Sample a batch with size $B$: $\{(\mathbf{s}_i, \mathbf{a}_i, r_i, \mathbf{s}_i')\}_{b=1}^B \sim \mathcal{D}$.
3: Shuffle batch $\{(\mathbf{s}_i, \mathbf{a}_i, r_i, \mathbf{s}_i')\}_{b=1}^B$ to $\{(\mathbf{s}_j, \mathbf{a}_j, r_j, \mathbf{s}_j')\}_{b=1}^B$.
4: Update Q network by (8) with DrQ augmentation.
5: update actor network by (11).
6: update alpha $\alpha$.
7: update encoder $\Theta$ by (10)
8: update dynamics model $\hat{\mathcal{P}}$.

---

a different parameter $\mathbf{v}_*^{(2)}$ except for the last term, where the parameter $\mathbf{v}_*^{(1)}$ does not change. The reason why the parameter $\mathbf{v}_*^{(1)}$ is used in the last term is that we expect the meta-learner $f_d$ is able to make a consensus prediction on $\mathbf{s}_*^{(1)}$ and $\mathbf{s}_*^{(2)}$ as they are transformed from a same state $\mathbf{s}_*$. Another major difference between the first two terms and the last two terms is the order of the subscripts $i$ and $j$: $i$ is followed by $j$ in first two terms while $j$ is followed by $i$ in last two terms. The reason behind this is that we aim to make $f_r$ and $f_d$ to be symmetric.

The loss of the actor of SAC, which is based on the output of the encoder, is

$$J_\pi(\psi) = \mathbb{E}_{(\mathbf{s},\mathbf{a})\sim\mathcal{D}}\left[\alpha \log(\pi_\psi(\mathbf{a}|\hat{\phi}(\mathbf{s}))) - Q_\theta(\phi(\mathbf{s}), \mathbf{a})\right], \tag{11}$$

where $\hat{\phi}$ denotes the convolutional layers of $\phi$ and $Q_\theta(\phi(\mathbf{s}), \mathbf{a})$ is a convenient form of the Q-function with $c$-weighted state representations input. We also stop gradients for $\hat{\phi}$ and $\phi$ ($\phi_r$ and $\phi_d$) regarding the actor loss. In other words, the loss in (11) only calculates the gradients w.r.t. $\psi$.

The overall learning algorithm is summarized in Algorithm 1. Note that due to limit of space, more details can be found in Appendix A. In summary, as illustrated in Figure 1, in AMBS, the convnet is shared by the encoders $\phi_r$ and $\phi_d$, which consists of 4 convolutional layers. The final output of

the convnet is fed into two branches: 1) one is a fully-connected layer to form reward representation $\phi_r(\mathbf{s})$, and 2) the other is also a fully-connected layer but forms dynamics representation $\phi_d(\mathbf{s})$. The meta-learners $f_r$ and $f_d$ are both MLPs with two layers. More implementation details are provided at Appendix B. Source code is available at `https://github.com/jianda-chen/AMBS`.

## 5 EXPERIMENTS

The major goal of our proposed AMBS is to learn a generalizable representation for RL from high-dimensional raw data. We evaluate AMBS on two environments: 1) the DeepMind Control (DMC) suite (Tassa et al., 2018) with background distraction; and 2) autonomous driving task on CARLA (Dosovitskiy et al., 2017) simulator. Several baselines methods are compared with AMBS: 1) Deep Bisimulation Control (DBC) (Zhang et al., 2021) which is recent research on RL representation learning by using the $L_1$ distance to approximate bisimulation metric; 2) PSEs (Agarwal et al., 2021) which proposes a new state-pair metric called policy similarity metric for representation learning and learns state embedding by incorporating a contrastive learning method SimCLR (Chen et al., 2020); 3) DrQ (Yarats et al., 2021), a state-of-the-art data augmentation method for deep RL; 4) Stochastic Latent Actor-Critic (SLAC) (Lee et al., 2020a), a state-of-the-art approach for partial observability on controlling by learning a sequential latent model with a reconstruction loss; 5) Soft Actor-Critic (SAC) (Haarnoja et al., 2018), a commonly used off-policy deep RL method, upon which above baselines and our approach are built.

### 5.1 DMC SUITE WITH BACKGROUND DISTRACTION

DeepMind Control (DMC) suite (Tassa et al., 2018) is an environment that provides high dimensional pixel observations for RL tasks. DMC suite with background distraction (Zhang et al., 2018; Stone et al., 2021) replaces the background with natural videos which are task-irrelevant to the RL tasks. We perform the comparison experiments in two settings: original background and nature video background. For each setting we evaluate AMBS and baselines in 4 environments, cartpole-swingup, finger-spin, cheetah-run and walker-walk. Results of additional environments are shown in Appendix D.

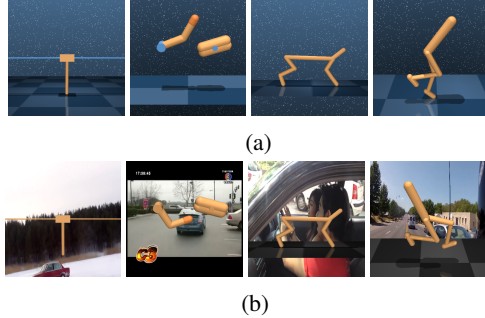

(a)

(b)

Figure 3: **(a)** Pixel observations of DMC suite with original background. **(b)** Pixel observations of DMC suite with natural video background. Videos are sampled from Kinetics (Kay et al., 2017).

**Original Background.** Figure 3a shows the DMC observation of original background which is static and clean. The experiment result is shown in Figure 4. Our method AMBS has comparable learning efficiency and converge scores comparing to state-of-the-art pixel-level RL methods DrQ and SLAC. Note that DBC performs worse in original background setting.

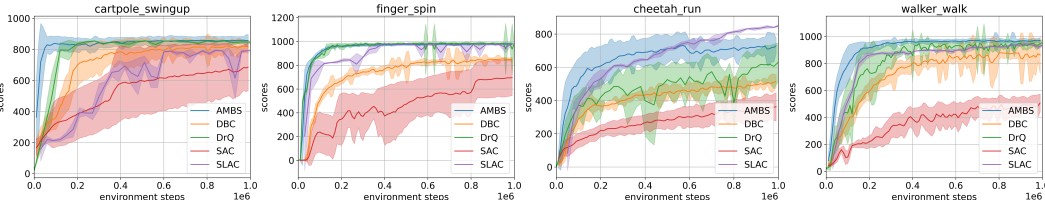

Figure 4: Training curves of AMBS and comparison methods. Each curve is average on 5 runs.

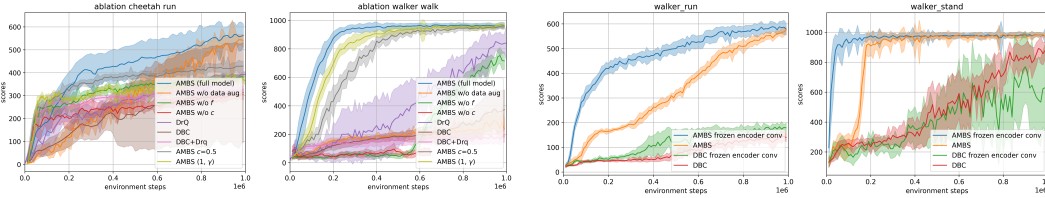

Figure 5: Training curves of AMBS and comparison methods on natural video background setting. Videos are sampled from Kinetics (Kay et al., 2017) dataset. Each curve is average on 5 runs.

**Natural Video Background.** Figure 3b shows the DMC observation with natural video background. The background video is sampled from the Kinetics (Kay et al., 2017) dataset under label "driving car". By following the experimental configuration of DBC, we sample 1,000 continuous frames as background video for training, and sample another 1,000 continuous frames for evaluation. The experimental result is shown in Figure 5. Our method AMBS outperforms other baselines in terms of both efficiency and scores. It converges to the scores that are similar to the scores on original background setting within 1 million steps. DrQ can only converge to the same scores on cartpole-swingup and finger-spin but the learning is slightly slower than AMBS. The sequential latent model method SLAC performs badly on this setting. This experiment demonstrates that AMBS is robust to the background task-irrelevant information and the AMBS objectives improve the performance when comparing to other RL representation learning methods, e.g., DBC.

## 5.2 ABLATION STUDY

Our approach AMBS consists of several major components: meta-learners, a factor to balance the impact between reward and dynamics, and data augmentation on representation learning. To evaluate the importance of each component, we eliminate or replace each component by baseline methods. We conduct experiments on cheetah-run and walker-walk under the natural video background setting. Figure 6 shows the performance of AMBS, variants of AMBS and baseline methods. **AMBS w/o data aug** is constructed by removing data augmentation in AMBS. **AMBS w/o** $f$ is referred to as replacing the meta-learners $f_r$ and $f_d$ by the $L_1$ distance, as in DBC. **AMBS w/o** $c$ removes the component of balancing impact between reward and dynamics but encodes the observation into a single embedding. It uses single meta-learner to approximate the sum of reward and dynamics distance. **DBC + DrQ** is DBC with data augmentation applied on representation learning and SAC. **AMBS** $c = 0.5$ denotes AMBS with a fixed weight $c = 0.5$. **AMBS w/** $(1, \gamma)$ replaces $c$ in AMBS by freezing the reward weight to 1 and the dynamics weight to $\gamma$. Such a setting of weights is used in DBC. Figure 6 demonstrates that AMBS performs better than any of its variant. Comparing to DBC that does not utilize data augmentation, our variant **AMBS w/o data aug** still performs better. This comparison shows that using meta-learners $f_r$ and $f_d$ to learn reward- and dynamics-relevant representations is able to improve RL performance compared with using the $L_1$ distance to approximate the whole $\pi$-bisimulation metric.

## 5.3 TRANSFER OVER REWARD FUNCTIONS

The environments walker-walk, walker-run and walker-stand share the same dynamics but have different reward functions according to the moving speed. To evaluate the transfer ability of AMBS over reward functions, we transfer the learned model from walker-walk to walker-run and walker-stand. We train agents on walker-run and walker-stand with frozen convolutional layers of encoders that are well trained in walker-walk. The experiments are done under the natural video setting

Figure 6: Ablation Study on cheetah-run **(left)** and walker-walk **(right)** in natural video setting.

Figure 7: Transfer from walker-walk to **(left)** walker-run and **(right)** walker-stand.

compared with DBC as shown in Figure 7. It shows that the transferring encoder converges faster than training from scratch. Besides our approach has a better transfer ability than DBC.

## 5.4 Generalization over Background Video

We evaluate the generalization ability of our method when background video is changed. We follow the experiment setting of PSE (Agarwal et al., 2021)n. We pick 2 videos, 'bear' and 'bmx-bumps', from the training set of DAVIS 2017 (Pont-Tuset et al., 2018) for training. We randomly sample one video and a frame at each episode start, and then play the video forward and backward until episode ends. The RL agent is evaluated on validation environment where the background video is sampled from validation set of DAVIS 2017. Those validation videos are unseen during the training. The evaluation scores at 500K environment interaction steps is shown in Table 1. The comparison method DrQ+PSEs is data augmented by DrQ. AMBS outperforms DrQ+PSEs on all the 4 tasks. It demonstrates that AMBS is generalizable on unseen background videos.

| Methods | C-swingup | F-spin | C-run | W-walk |
|---|---|---|---|---|
| DrQ + PSEs | 749 ± 19 | 779 ± 49 | 308 ± 12 | 789 ± 28 |
| AMBS | **807 ± 41** | **933 ± 96** | **332 ± 27** | **893 ± 85** |

Table 1: Generalization to unseen background videos sampled from DAVIS 2017 (Pont-Tuset et al., 2018). Scores of DrQ+PSEs are reported from Agarwal et al. (2021).

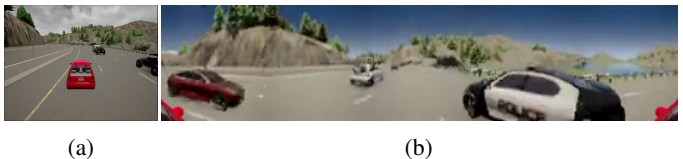

(a)          (b)

Figure 8: **(a)** Illustration of a third-person view in "Town4" scenario of CARLA. **(b)** A first-person observation for RL agent. It concatenates five cameras for 300 degrees view .

## 5.5 Autonomous Driving Task on CARLA

CARLA is an autonomous driving simulator that provides a 3D simulation for realistic on-road scenario. In real world cases or in realistic simulation, the learning of RL agents may suffer from complex background that contains task-irrelevant details. To argue that AMBS can address this issue, we perform experiments with CARLA. We follow the setting of DBC to create an autonomous driving task on map "Town04" for controlling a car to drive as far as possible in 1,000 frames. The environment

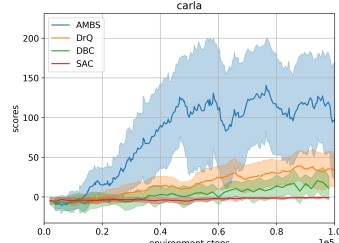

Figure 9: CARLA simulation.

contains a highway with 20 moving vehicles. The reward function prompts the driving distance and penalizes collisions. The observation shape is 84 × 420 pixels which consists of five 84 × 84 cameras. Each camera has 60 degrees view, together they produce a 300 degrees first-person view, as shown in Figure 8. The weather, clouds and brightness may change slowly during the simulation. Figure 9 shows the training curves of AMBS and other comparison methods. AMBS performs the best, which provides empirical evidence that AMBS can potentially generalize over task-irrelevant details, e.g., weather and clouds, in real world scenarios.

## 6 Conclusion

This paper presents a novel framework AMBS with meta-learners for learning task-relevant and environment-detail-invariant state representation for deep RL. In AMBS, we propose state encoders that decompose an observation into rewards and dynamics representations for measuring behavioral similarities by two self-learned meta-learners, and design a learnable strategy to balance the two types of representations. AMBS archives new state-of-the-art results in various environments and demonstrates its transfer ability on RL tasks. Experimental results verify that our framework is able to effectively learn a generalizable state representation.

ACKNOWLEDGEMENT

This work is supported by Microsoft Research Asia collaborative research grant 2020.

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

## A OBJECTIVES OF ADAPTIVE META-LEARNER OF BEHAVIORAL SIMILARITIES (AMBS)

We denote two encoders $\phi_r$ and $\phi_d$, where each encoder maps a high-dimensional observation to a low-dimensional representation. The encoder $\phi_r : \mathcal{S} \to \mathcal{Z}_r$ captures reward-relevant features and the encoder $\phi_d : \mathcal{S} \to \mathcal{Z}_d$ captures dynamics-relevant features, where $\mathcal{Z}_r$ and $\mathcal{Z}_d$ are representation spaces. We design two meta-learners $f_r : \mathcal{Z}_r \times \mathcal{Z}_r \to \mathbb{R}$ and $f_d : \mathcal{Z}_d \times \mathcal{Z}_d \to \mathbb{R}$ to output reward difference and dynamics difference, respectively. We follow DrQ (Yarats et al., 2021) to define the state transformation $h : \mathcal{S} \times \mathcal{T} \to \mathcal{S}$ that maps a state to a data-augmented state, where $\mathcal{T}$ is the space of parameters of $h$. In practice, we use random crop as transformation $h$.

With augmentation transformation $h$, the loss of representation learning is,

$$
\begin{aligned}
\ell(\Theta) =& (1-c)\left( f_r(\phi_r(\mathbf{s}_i^{(1)}), \phi_r(\mathbf{s}_j^{(1)})) - |r_i - r_j| \right)^2 \\
&+ c\left( f_d(\phi_d(\mathbf{s}_i^{(1)}), \phi_d(\mathbf{s}_j^{(1)})) - W_2(\hat{\mathcal{P}}(\cdot|\phi_d(\mathbf{s}_i^{(1)}), \mathbf{a}_i), \hat{\mathcal{P}}(\cdot|\phi_d(\mathbf{s}_j^{(1)}), \mathbf{a}_j)) \right)^2 \\
&+ (1-c)\left( f_r(\phi_r(\mathbf{s}_j^{(2)}), \phi_r(\mathbf{s}_i^{(2)})) - |r_i - r_j| \right)^2 \\
&+ c\left( f_d(\phi_d(\mathbf{s}_j^{(2)}), \phi_d(\mathbf{s}_j^{(2)})) - W_2(\hat{\mathcal{P}}(\cdot|\phi_d(\mathbf{s}_i^{(1)}), \mathbf{a}_i), \hat{\mathcal{P}}(\cdot|\phi_d(\mathbf{s}_j^{(1)}), \mathbf{a}_j)) \right)^2,
\end{aligned}
\tag{12}
$$

where $\Theta = \{f_r, f_d, \phi_r, \phi_d\}$, $\mathbf{s}_*^{(l)} = h(\mathbf{s}_*, \mathbf{v}_*^{(l)})$ is the transformed state with parameters $\mathbf{v}_*^{(l)}$. Specifically, $\mathbf{s}_i^{(1)}$, $\mathbf{s}_j^{(1)}$, $\mathbf{s}_i^{(2)}$ and $\mathbf{s}_j^{(2)}$ are transformed states with parameters of $\mathbf{v}_i^{(1)}$, $\mathbf{v}_j^{(1)}$, $\mathbf{v}_i^{(2)}$ and $\mathbf{v}_j^{(2)}$ respectively, which are all drawn from $\mathcal{T}$ independently. $\hat{\mathcal{P}}(\cdot|\phi_d(\mathbf{s}_*), \mathbf{a}_*)$ is a learned probabilistic dynamics model, which outputs a Gaussian distribution over $\mathcal{Z}_d$ for predicting dynamics representation of next state $\mathbf{s}_*'$. $W_2$ denotes the 2-Wasserstein distance which has closed-form for Gaussian distributions: $W_2(\mathcal{N}(\mu_i, \boldsymbol{\Sigma}_i), \mathcal{N}(\mu_j, \boldsymbol{\Sigma}_j))^2 = \|\mu_i - \mu_j\|_2^2 + \|\boldsymbol{\Sigma}_i^{\frac{1}{2}} - \boldsymbol{\Sigma}_j^{\frac{1}{2}}\|_{\mathcal{F}}^2$, where $\mu_* \in \mathbb{R}^{|\mathcal{Z}_d|}$ are the mean vectors of Gaussian distribution, $\boldsymbol{\Sigma}_* \in \mathbb{R}^{|\mathcal{Z}_d|}$ are the diagonals of covariance matrices, and $\|\cdot\|_{\mathcal{F}}$ is Frobenius norm. Note that we stop gradients for $c$ as mentioned in Section 4.2.

The adaptive weight $c$ and $(1-c)$ are the softmax output of corresponding learnable parameter $\eta_c \in \mathbb{R}^2$. It can be formulated as

$$
[c : (1-c)] = softmax(\eta_c).
\tag{13}
$$

where the operation : denotes concatenation of two vectors/scalars,

We integrate the learning of $c$ into the update of Q-function of SAC. The objective of the Q-function with DrQ augmentation and clipped double trick is

$$
\begin{aligned}
J_Q(\theta_k, \eta_c) = \mathbb{E}_{(\mathbf{s}, \mathbf{a}, \mathbf{s}') \sim \mathcal{D}} \bigg[ &\frac{1}{2}\left( Q_{\theta_k}([(1-c)\phi_r(\mathbf{s}^{(1)}) : c\phi_d(\mathbf{s}^{(1)})], \mathbf{a}) - (r(\mathbf{s}, \mathbf{a}) + \gamma V(\mathbf{s}')) \right)^2 \\
&+ \frac{1}{2}\left( Q_{\theta_k}([(1-c)\phi_r(\mathbf{s}^{(2)}) : c\phi_d(\mathbf{s}^{(2)})], \mathbf{a}) - (r(\mathbf{s}, \mathbf{a}) + \gamma V(\mathbf{s}')) \right)^2 \bigg],
\end{aligned}
\tag{14}
$$

where $\mathbf{s}^{(l)} = h(\mathbf{s}, \mathbf{v}^{(l)})$, $\mathbf{v}^{(l)}$ are all drawn from $\mathcal{T}$ independently, and $\{Q_{\theta_k}\}_{k=1}^2$ are the double Q networks. The target value function with DrQ augmentation is

$$
\begin{aligned}
V(\mathbf{s}') = \frac{1}{2}\bigg( &\min_{k=1,2} Q_{\bar{\theta}_k}\left( [(1-\bar{c})\bar{\phi}_r(\mathbf{s}'^{(1)}) : \bar{c}\bar{\phi}_d(\mathbf{s}'^{(1)})], \mathbf{a}'^{(1)} \right) + \alpha \log \pi_\psi(\mathbf{a}'^{(1)}|\mathbf{s}'^{(1)}) \\
&+ \min_{k=1,2} Q_{\bar{\theta}_k}\left( [(1-\bar{c})\bar{\phi}_r(\mathbf{s}'^{(2)}) : \bar{c}\bar{\phi}_d(\mathbf{s}'^{(2)})], \mathbf{a}'^{(2)} \right) + \alpha \log \pi_\psi(\mathbf{a}'^{(2)}|\mathbf{s}'^{(2)}) \bigg),
\end{aligned}
\tag{15}
$$

where $\mathbf{s}'^{(l)} = h(\mathbf{s}', \mathbf{v}'^{(l)})$, $\mathbf{a}'^{(l)} \sim \pi_\psi(\mathbf{a}'^{(l)}|\mathbf{s}'^{(l)})$, $\bar{\theta}_k$ is the set of parameters of the target Q network, $\bar{\phi}_r$ and $\bar{\phi}_d$ are target encoders specifically for the target Q network and $\bar{c}$ is the adaptive weight for target encoders. The parameters $\bar{\theta}_k$, $\bar{\phi}_r$, $\bar{\phi}_d$ and $\bar{\eta}_c$ are softly updated.

The loss of actor of SAC is,

$$
J_\pi(\psi) = \mathbb{E}_{(\mathbf{s}) \sim \mathcal{D}}\left[ \mathbb{E}_{\mathbf{a} \sim \pi_\psi}[\alpha \log(\pi_\psi(\mathbf{a}|\hat{\phi}(\mathbf{s}^{(1)}))) - \min_{k=1,2} Q_{\theta_k}(\phi(\mathbf{s}^{(1)}), \mathbf{a})] \right].
\tag{16}
$$

The loss of $\alpha$ of SAC is,

$$J(\alpha) = \mathbb{E}_{(\mathbf{s}) \sim \mathcal{D}} \left[ \mathbb{E}_{\mathbf{a} \sim \pi_\psi} [\alpha \log(\pi_\psi(\mathbf{a}|\hat{\phi}(\mathbf{s}^{(1)}))) - \alpha \bar{\mathcal{H}}] \right], \tag{17}$$

where $\bar{\mathcal{H}} \in \mathbb{R}$ is the target entropy hyper-parameter which in practice is $\bar{\mathcal{H}} = -|\mathcal{A}|$.

The loss for updating of dynamics model $\hat{\mathcal{P}}$ is

$$\ell(\hat{\mathcal{P}}) = \mathbb{E}_{(\mathbf{s},\mathbf{a},\mathbf{s}') \sim \mathcal{D}} \left[ \left( \frac{\phi_d(\mathbf{s}'^{(1)}) - \mu(\hat{\mathcal{P}}(\cdot|\phi_d(\mathbf{s}^{(1)}), \mathbf{a}))}{2\sigma(\hat{\mathcal{P}}(\cdot|\phi_d(\mathbf{s}^{(1)}), \mathbf{a}))} \right)^2 \right], \tag{18}$$

where $\mu(\hat{\mathcal{P}}(\cdot))$ and $\sigma(\hat{\mathcal{P}}(\cdot))$ are mean and standard deviation of $\hat{\mathcal{P}}$ output Gaussian distribution, respectively.

Algorithm 2 shows the algorithm at each learning step.

---

**Algorithm 2** AMBS+SAC

---

1: **Input:** Replay Buffer $\mathcal{D}$, initialized $\Theta = \{f_r, \phi_r, f_d, \phi_d\}$, Q network $Q_{\theta_k}$, actor $pi_\psi$, target Q network $Q_{\bar{\theta}_k}$,
2: Sample a batch with size $B$: $\{(\mathbf{s}_i, \mathbf{a}_i, r_i, \mathbf{s}'_i)\}_{b=1}^{B} \sim \mathcal{D}$.
3: Shuffle batch $\{(\mathbf{s}_i, \mathbf{a}_i, r_i, \mathbf{s}'_i)\}_{b=1}^{B}$ to $\{(\mathbf{s}_j, \mathbf{a}_j, r_j, \mathbf{s}'_j)\}_{b=1}^{B}$.
4: Update Q network by equation 14 with DrQ augmentation.
5: Update actor network by equation 16 .
6: Update alpha $\alpha$ by equation 17 .
7: Update encoder $\Theta$ by equation 12
8: Update dynamics model $\hat{\mathcal{P}}$ by equation 18.
9: Softly update target Q network: $\bar{\theta}_k = \tau_Q \theta_k + (1 - \tau_Q)\bar{\theta}_k$ .
10: Softly update target encoder: $\bar{\phi} = \tau_\phi \phi + (1 - \tau_\phi)\bar{\phi}$ .
11: Softly update target $\bar{c}$: $\bar{\eta}_c = \tau_\phi \eta_c + (1 - \tau_\phi)\bar{\eta}_c$ .

---

# B IMPLEMENTATION DETAILS

## B.1 PIXELS PROCESSING

We stack 3 consecutive frames as observation, where each frame is $84 \times 84$ RGB images in DeepMind control suite. Each pixel is divided by 255 to down scale to $[0, 1]$. We consider the stacked frames as fully observed state.

## B.2 NETWORK ARCHITECTURE

We share the convnet for encoder $\phi_r$ and encoder $\phi_d$. The shared convnet consists of four convolutional layers with $3 \times 3$ kernels and 32 output channels. We set stride to 1 everywhere, except for the first convolutional layer, which has stride 2. We use ReLU activation for each convolutional layer output. The final output of convnet is fed into two branches: 1) one is a fully-connected layer with 50 dimensions to form reward representation $\phi_r(\mathbf{s})$, and 2) the other one is also a fully-connected layer with 50 dimensions but form dynamics representation $\phi_d(\mathbf{s})$. The meta-learners $f_r$ and $f_d$ are both MLPs with two layers with 50 hidden dimensions.

The critic network takes $\phi_r(\mathbf{s})$, $\phi_d(\mathbf{s})$ and action as input, and feeds them into three stacked fully-connected layers with 1024 hidden dimensions. The actor network takes the shared convnet output as input, and feeds it into four fully-connected layers where the first layer has 100 hidden dimensions and other layers have 1024 hidden dimensions. The dynamics model is MLPs with two layers with 512 hidden dimensions. ReLU activations are used in every hidden layer.

## B.3 HYPERPARAMETERS

The hyperparameters used in our algorithm are listed in Table 2.

| Parameter name | Value |
|---|---|
| Replay buffer capacity | $10^6$ |
| Discount factor $\gamma$ | 0.99 |
| Minibatch size | 128 |
| Optimizer | Adam |
| Learning rate for $\alpha$ | $10^{-4}$ |
| Learning rate except $\alpha$ | $5 \times 10^{-4}$ |
| Target update frequency | 2 |
| Actor update frequency | 2 |
| Actor log stddev bounds | $[-10, 2]$ |
| $\tau_Q$ | 0.01 |
| $\tau_\phi$ | 0.05 |
| Init temperature | 0.1 |

Table 2: Hyperparameters used in our algorithm.

## B.4 ACTION REPEAT FOR DMC

We use different action repeat hyper-parameters for each task in DeepMind control suite, which are listed in Table 3.

| Task name | Action repeat |
|---|---|
| Cartpole Swingup | 8 |
| Finger Spin | 2 |
| Cheetah Run | 4 |
| Walker Walk | 2 |
| Reacher Easy | 4 |
| Ball In Cup Catch | 4 |

Table 3: Action repeat used for each task in DeepMind control suite.

## B.5 SOURCE CODES

Source code of AMBS is available at `https://github.com/jianda-chen/AMBS`.

## C ANALYSIS

### C.1 REGRESSION LOSSES OF APPROXIMATING BISIMULATION METRIC

We compare the regression losses of approximating the bisimulation metric over RL environment steps among DBC and our AMBS. DBC uses the $L_1$ norm to calculate the distances between two states embeddings. AMBS performs meta-learners on state representations to measure the distance with learned similarities. Figure 10 demonstrates that AMBS has smaller regression loss and also faster descent tendency compared to DBC. Meta-learners in AMBS are able to provide more stable gradients to update the state representations.

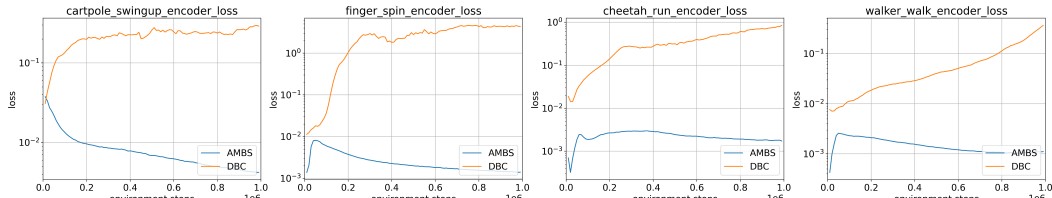

Figure 10: The regression losses over RL environment steps among DBC and AMBS.

## C.2 VISUALIZATIONS OF COMBINATION WEIGHT c

Figure 11 shows the values of combination weight $c$ in AMBS trained on DMC with origin background and Figure 11 is on DMC with video background setting. The values of c at 1M steps vary in different tasks. The common trend is that they increase at the beginning of training. In original background setting $c$ changes slowly after the beginning. In natural video background setting , $c$ has a fast drop after the beginning. In the beginning, agent learns a large weight $c$ for dynamics feature that may boost the learning of RL agent. Then it drops for natural video background setting because it learns that the video background is distracting the agent. Lower weight is learned for the dynamics features.

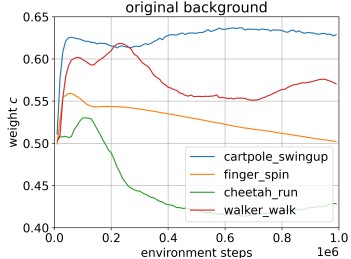

Figure 11: The value of combination weight $c$ over environment steps. AMBS is trained on DMC on Original background.

Figure 12: The value of combination weight $c$ over environment steps. AMBS is trained on DMC on video background.

## C.3 NORMS OF STATE EMBEDDINGS

We record the $L_1$ norms of reward representation $\phi_r(s)$ and dynamics representation $\phi_d(s)$ during training. We record $\frac{1}{n_r}||\phi_r(s)||_1$ for reward representation where $n_r$ is the number of dimensions of $\phi_r(s)$, and $\frac{1}{n_d}||\phi_d(s)||_1$ for dynamics representation where $n_r$ is the number of dimensions of $\phi_d(s)$. Figure 13 shows the values averaged on each sampled training batch. Although the $L_1$ norm of $\phi_d(s)$ decreases during training, it converges around 0.4-0.5. The $L_1$ norm of $\phi_r(s)$ also decreases in the training procedure. It can verify that dynamics representation $\phi_d(s)$ will not converge to all zeros.

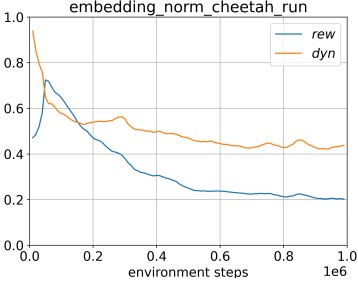

Figure 13: Norms of state embeddings: $\frac{1}{n_r}||\phi_r(s)||_1$ and $\frac{1}{n_d}||\phi_d(s)||_1$.

## C.4 Sharing Encoders Between Actor and Critic

In AMBS, we share only the convolutional layers $\hat{\phi}$ of encoders between actor and critic network. We compare to sharing the whole encoders $\phi_r$ and $\phi_d$ in this experiment. Figure 14 shows the raining curves on Walker-Walk with natural video background. Sharing full encoder $\phi_r$ and $\phi_d$ is worse than sharing only CNN $\hat{\phi}$.

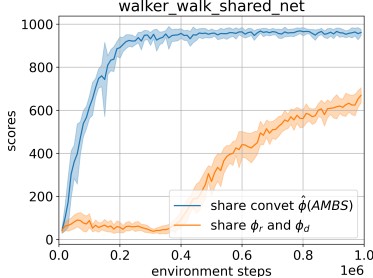

Figure 14: Training curves of different ways of sharing encoder. Experiments on Walker-Walk with natural video background.

## D Additional Experimental Results

### D.1 Additional Result of DMC Suite with and w/o Background Distraction

Figure 15 shows the training curves on original background setting. Figure 16 shows the training curves on Kinetics (Kay et al., 2017) video background setting.

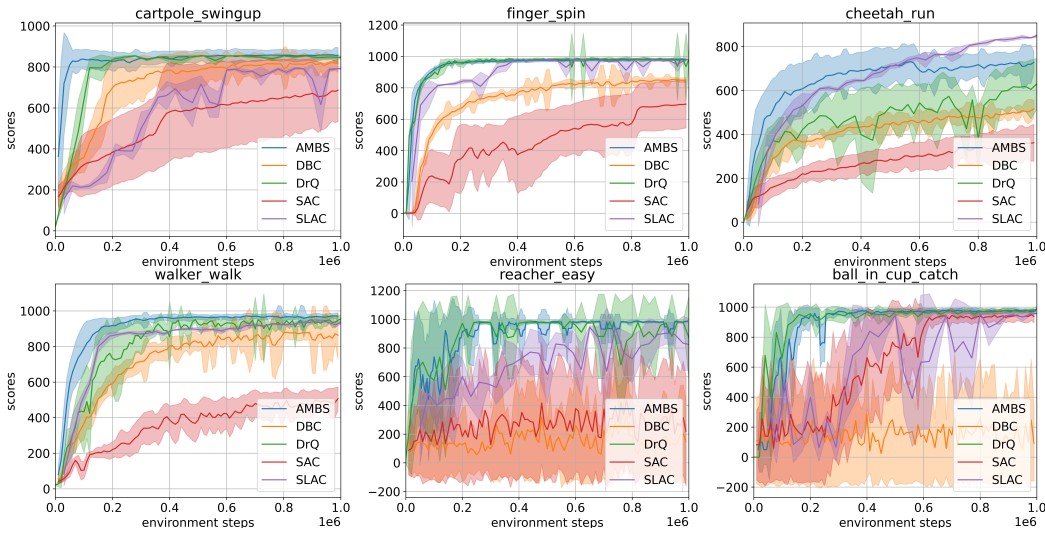

Figure 15: Training curves of AMBS and comparison methods. Each curve is average on 3 runs.

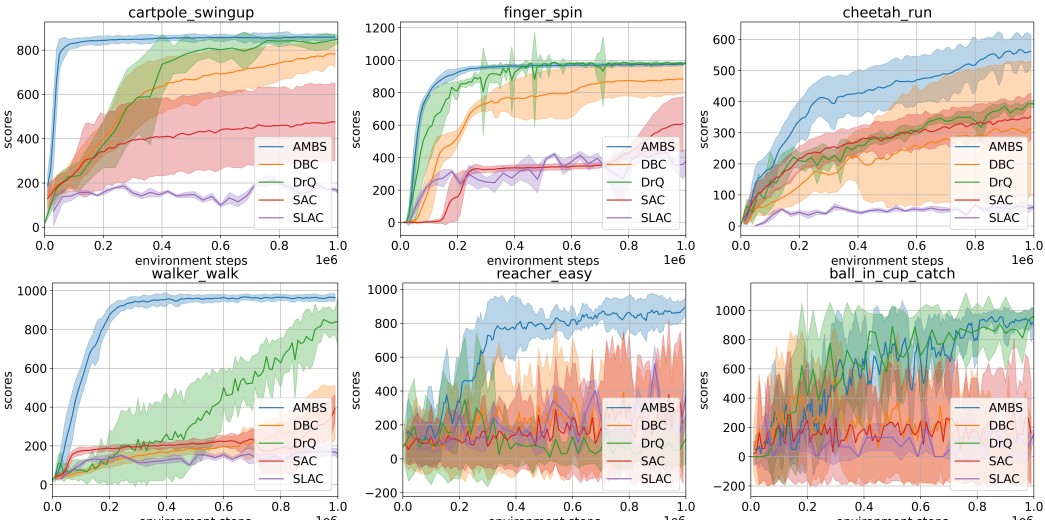

Figure 16: Training curves of AMBS and comparison methods on natural video background setting. Videos are sampled from Kinetics (Kay et al., 2017) dataset. Each curve is average on 3 runs.

## D.2 ADDITIONAL RESULT OF GENERALIZATION OVER BACKGROUND VIDEO

Table 4 shows the scores on DAVIS video background setting comparing to PSEs (Agarwal et al., 2021).

| Methods | C-swingup | F-spin | C-run | W-walk | R-easy | BiC-catch |
|---|---|---|---|---|---|---|
| DrQ + PSEs | $749 \pm 19$ | $779 \pm 49$ | $308 \pm 12$ | $789 \pm 28$ | $955 \pm 10$ | $821 \pm 17$ |
| AMBS | $\mathbf{807 \pm 41}$ | $\mathbf{933 \pm 96}$ | $\mathbf{332 \pm 27}$ | $\mathbf{893 \pm 85}$ | $\mathbf{980 \pm 11}$ | $\mathbf{944 \pm 59}$ |

Table 4: Generalization with unseen background videos sampled from DAVIS 2017 (Pont-Tuset et al., 2018) dataset. We evaluate the agents at 500K environment steps. Scores of DrQ + PSEs are reported from Agarwal et al. (2021).

## D.3 ADDITIONAL RESULT ON CARLA

The CARLA experiment is to learn an autonomous driving RL agent to avoid collisions with other moving vehicles and drive as far as possible on a highway within 1000 frames. Those vehicles are randomly generated, including vehicles' types, colors and initial positions. We raise the number of generated vehicles from 20 to 40 so that the road becomes more crowded for RL agent. Besides, we also extend the training steps to 2e5 for AMBS and DBC. Figure 17 shows the training curve on CARLA. **AMBS** and **DBC** run on the precious setting of CARLA, while **AMBS40** and **DBC40** run on the highway with 40 generated vehicles. When increasing the number of vehicles from 20 to 40, DBC becomes more difficult to learn but AMBS is just slightly slower. DBC still doesn't achieve high scores after 1e5 steps.

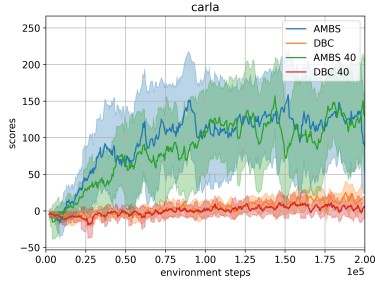

Figure 17: Training curve on CARLA environment.

# E  PROOFS

## E.1  VALUE-FUNCTION BOUND

**Theorem E.1** (Value function difference bound for different discounting factors (Petrik & Scherrer, 2009; Kemertas & Aumentado-Armstrong, 2021)). *Consider two identical MDPs except for different discounting factors $\gamma_1$ and $\gamma_2$, where $\gamma_1 \leq \gamma_2$ and reward $r \in [0, 1]$. Let $V_\gamma^\pi$ denotes the value function for MDP with discounting factor $\gamma$ given policy $\pi$.*

$$\left| V_{\gamma_1}^\pi(\mathbf{s}) - V_{\gamma_2}^\pi(\mathbf{s}) \right| \leq \frac{\gamma_2 - \gamma_1}{(1 - \gamma_1)(1 - \gamma_2)} \tag{19}$$

**Definition E.1** (*c*-weighted on policy bisimulation metric). *Given a policy $\pi$, and $c \in (0, 1)$, the on-policy bisimulation metric exists,*

$$d(\mathbf{s}_i, \mathbf{s}_j) = (1 - c)|\mathcal{R}_{\mathbf{s}_i}^\pi - \mathcal{R}_{\mathbf{s}_j}^\pi| + cW_1(d)(\mathcal{P}_{\mathbf{s}_i}^\pi, \mathcal{P}_{\mathbf{s}_j}^\pi). \tag{20}$$

**Theorem E.2** (Value difference bound). *For any $c \in [\gamma, 1)$, given two state $\mathbf{s}_i$ and $\mathbf{s}_j$,*

$$(1 - c)\left|V^\pi(\mathbf{s}_i) - V^\pi(\mathbf{s}_j)\right| \leq d(\mathbf{s}_i, \mathbf{s}_j) \tag{21}$$

*proof.* Lemma 6 of Kemertas & Aumentado-Armstrong (2021).

**Theorem E.3** (Generalized value difference bound). *For any $c \in (0, 1)$, given two states $\mathbf{s}_i$ and $\mathbf{s}_j$,*

$$(1 - c)|V^\pi(\mathbf{s}_i) - V^\pi(\mathbf{s}_j)| \leq d(\mathbf{s}_i, \mathbf{s}_j) + \frac{2(1 - c)(\gamma - \min(c, \gamma))}{(1 - \gamma)(1 - c)} \tag{22}$$

*proof.* We follow the proof of Theorem 1 in Kemertas & Aumentado-Armstrong (2021). Suppose another MDP with discounting factor $\gamma' = c$ exists. From Theorem E.2 we have

$$(1 - c)|V_{\gamma'}^\pi(\mathbf{s}_i) - V_{\gamma'}^\pi(\mathbf{s}_j)| \leq d(\mathbf{s}_i, \mathbf{s}_j). \tag{23}$$

Then,

$$
\begin{aligned}
&(1 - c)|V^\pi(\mathbf{s}_i) - V^\pi(\mathbf{s}_j)| \\
=&(1 - c)|V^\pi(\mathbf{s}_i) - V_{\gamma'}^\pi(\mathbf{s}_i) + V_{\gamma'}^\pi(\mathbf{s}_i) - V^\pi(\mathbf{s}_j) + V_{\gamma'}^\pi(\mathbf{s}_j) - V_{\gamma'}^\pi(\mathbf{s}_j)| \\
\leq&(1 - c)(|V_{\gamma'}^\pi(\mathbf{s}_i) - V_{\gamma'}^\pi(\mathbf{s}_j)| + |V^\pi(\mathbf{s}_i) - V_{\gamma'}^\pi(\mathbf{s}_i)| + |V^\pi(\mathbf{s}_j) - V_{\gamma'}^\pi(\mathbf{s}_j)|) \\
\leq&d(\mathbf{s}_i, \mathbf{s}_j) + (1 - c)(|V^\pi(\mathbf{s}_i) - V_{\gamma'}^\pi(\mathbf{s}_i)| + |V^\pi(\mathbf{s}_j) - V_{\gamma'}^\pi(\mathbf{s}_j)|) \\
\leq&d(\mathbf{s}_i, \mathbf{s}_j) + \frac{2(1 - c)(\gamma - \min(\gamma', \gamma))}{(1 - \gamma)(1 - \gamma')}
\end{aligned} \tag{24}
$$

Replace $\gamma'$ by $c$, finally we have

$$(1 - c)|V^\pi(\mathbf{s}_i) - V^\pi(\mathbf{s}_j)| \leq d(\mathbf{s}_i, \mathbf{s}_j) + \frac{2(1 - c)(\gamma - \min(c, \gamma))}{(1 - \gamma)(1 - c)}. \tag{25}$$

