# OpenReview forum: "Learning Generalizable Representations for Reinforcement Learning via Adaptive Meta-learner of Behavioral Similarities"
_ICLR.cc/2022/Conference — ICLR 2022 Poster_

### Official Review · Reviewer_bfCF · 2021-10-30

**Correctness:** 3
**Technical Novelty And Significance:** 2
**Empirical Novelty And Significance:** 3
**Recommendation:** 6
**Confidence:** 3

**Main Review:**


Strengths:
- The paper’s improvements over prior work are generally pretty reasonable and backed by convincing experimental results
- The experimental results are generally pretty thorough with relevant comparisons to prior works and ablations
- The writing is clear and exposition of the literature is thorough

Weaknesses:
- This paper generally feels like a “DBC with more bells and whistles” that enable it to work better; I think the general novelty is on the weaker side. The authors can consider a more thorough investigation into where existing representation learning methods fall short and address them more fundamentally
- The experiments with natural video backgrounds generally feel pretty contrived (granted this is what’s in the literature). The only realistic setting is CARLA and I would have liked to see more environments of that nature or more detailed experimentation there.

Detailed questions:
- If reward and dynamics have different encoders, to what extent is this still learning a bisimulation, or mapping to an equivalent state space?
- What do the experimental evaluations look like with more environment steps? Are the asymptotic performances of all methods similar or is AMBS just more sample efficient in the beginning? e.g. DBC uses 1e6 steps on CARLA
- nit: it’s not really clear to me the naming of “meta-learner”, what is the meta train/test tasks here? I think something like learned distance metric would be clearer



**Summary Of The Paper:**


This paper proposes a representation learning method for reinforcement learning via the bisimulation metric. It’s largely an extension of the work in Deep Bisimulation for Control (DBC) (Zhang et al., 2020) with some additional improvements:
- Learned distance metrics (“meta-learners”) instead of L1 distance on the learned representations
- Separate feature encodings for state and dynamics
- Data augmentation from DrQ (Yarats et al., 2021)
- Learned c tradeoff between reward and dynamics



**Summary Of The Review:**

Overall this paper’s contributions are generally reasonable and empirically backed, however its novelty and experimental evaluation leave much to be desired. To this end, I am leaning towards acceptance but would not be upset if this paper is rejected.

---

> ### Author Response · Authors · 2021-11-20
> **Response to Reviewer bfCF (1/2)**
>
> Thanks for your valuable comments and helpful suggestion!
>
> **Review4.1**
> This paper generally feels like a “DBC with more bells and whistles” that enable it to work better; I think the general novelty is on the weaker side.
>
> **Response:**
> In this manuscript, we propose a framework that learns behavioral similarities and learn generalizable state representations for deep RL. We observed that DBC suffers from the issues of approximating the bisimulation metric by L1 distance, then we propose AMBS using two neural network-based meta-learners, which are able to learn behavioral similarities faster and achieve lower approximation errors. Besides, we use two state encoders that learn two separate state embeddings regarding reward and dynamic, respectively, and design a learnable adaptive strategy to balance them, while DBC learns a single representation for each state.
>
>
> **Review4.2**
> The authors can consider a more thorough investigation into where existing representation learning methods fall short and address them more fundamentally.
>
> **Response:**
> Thanks for your suggestion.
> Previous representation learning methods in pixel-input RL rely on reconstruction loss. Reconstruction loss is computed over all the pixels, which results in all the details of the high-dimensional images being preserved in low-dimensional representations. It may overfit on the details and may not generalize well in complex environments. DBC is a reconstruction-free method and it evaluate bisimluation metric by L1 distance of states representations. DBC gets rid of overfitting on pixels but L1 distance is not easy to be optimized. Agarwal et al., (2021) introduce a variant of bisimulation metric replacing the reward distance by the difference of policy. They evaluate the state distance as cosine distance between representations and learn representation by a contrastive learning objective. But their method requires dynamic is deterministic.
>
>
> **Review4.3**
> The experiments with natural video backgrounds generally feel pretty contrived (granted this is what’s in the literature). The only realistic setting is CARLA and I would have liked to see more environments of that nature or more detailed experimentation there.
>
> **Response:**
> The natural video background DeepMind Control Suite is a benchmark environment for deep RL generalization. We evaluate our AMBS on such environment to make fair comparison to the current state-of-the-art methods (Zhang et al., 2020,  Agarwal et al., 2021), which also perform experiments on it.
> Besides, we would like to show more experimental results on CARLA. The CARLA experiment is to learn an autonomous driving RL agent to avoid collisions with other moving vehicles and drive as far as possible on a highway within 1000 frames. Those vehicles are randomly generated,  including vehicles' types, colors and initial positions. We increase the number of vehicles from 20 to 40, in order to make the roads more crowded and more difficult for the RL agent. The learning curves of this additional experiment are shown in Figure 16, Appendix D.3. Figure 16 shows that our AMBS still outperforms the comparison method DBC.
>
> **Review4.4**
> If reward and dynamics have different encoders, to what extent is this still learning a bisimulation, or mapping to an equivalent state space?
>
> **Response:**
> If we sum up the two learned metric $f_r$ and $f_d$ with weight $c$ to form $(1-c)f_r(\phi_r(s_i), \phi_r(s_j)) + c f_d(\phi_d(s_i), \phi_d(s_j))$, such metric can partially recover the bisimulation metric in DBC. The major difference is that we use learned $f_r$ and $f_d$ instead of L1 distance. If we learn separate representation but still using L1 distance, then concatenating two representations $[(1-c)\phi_r(s):c\phi_d(s)]$ can map to similar space as DBC.
>
> **Review4.5**
> What do the experimental evaluations look like with more environment steps? Are the asymptotic performances of all methods similar or is AMBS just more sample efficient in the beginning? e.g. DBC uses 1e6 steps on CARLA
>
> **Response:**
> We show 10e5 steps for CARLA because we want to make fair comparison with DBC(Zhang et al., 2020), which also shows only 1e5 steps. We extend the learning steps to 2e5 steps (due to the limited capacity of our machine, we cannot perform 1e6 steps in this period) and show the learning curves in Figure 16, Appendix D.3. Our AMBS still outperform the comparison method DBC within 2e5 steps.

---

> > ### Comment · Reviewer_bfCF · 2021-11-30
> > **Thank you**
> >
> > Thanks for the author's responses. I am still not fully convinced of the bisimulation argument with different encoders (see Reviewer Np3q's comments). This paper has decent experimental results, but is not theoretically or conceptually novel. I am leaning towards accept but would not be upset if this paper were rejected.

---

> > > ### Author Response · Authors · 2021-11-30
> > > **Re: Thank you**
> > >
> > > We have replied to Reviewer Np3q regarding the two different encoders. Here is for your information:
> > >
> > > - We discuss about the reason that we utilize two embeddings.
> > > Bisimulation metric uses hyper-parameters for linearly combining the reward and dynamic components. For example, DBC and Castro (2020) use 1 and $\gamma$ as the weights, and Ferns et al., (2004) uses $c_R$ and $c_T$. All of them manually assign the values for the linear combination. However we argue that such combination weights should vary in different environments. Meanwhile, Kemertas et al. (2021) provides a value bond for bisimulation metric that no more requires the restriction between combination weights and $\gamma$. Thus we aim to propose learning-based combination weights. In order to learn the combination weight $c$, we learn two embeddings ($\phi_r(s)$ and $\phi_d(s)$) then concatenate $\phi_r(s)$ and $\phi_d(s)$ weighted by $c$ and fed the concatenation into Q-function. In this way, $c$ can be optimized under Q-function loss. Otherwise, with single embedding (as how DBC does) there is no elegant ways to learn such $c$.
> > >
> > > - It is not true that $\phi_d$ captures nothing about rewards and $\phi_r$ captures nothing about dynamics. Embeddings $\phi_r(s)$ and $\phi_d(s)$ are concatenated and fed into Q-function. They are also optimized in Q-function loss (Equation (8)). Because Q-value is the expectation of future rewards, $\phi_r(s)$ and $\phi_d(s)$ somehow learn both rewards and future dynamics. Equation (6) encourages $\phi_r$ to capture more about the immediate rewards and Equation (7) encourages $\phi_d$ to capture more about the dynamics. Besides, $\phi_r(s)$ and $\phi_d(s)$ share convolutional layers in our implementation, as mentioned in Section 4.4 and Appendix B.2. The update of $\phi_r(s)$ and $\phi_d(s)$ can influence each other.

---

> ### Author Response · Authors · 2021-11-20
> **Response to Reviewer bfCF (2/2)**
>
> **Review4.6**
> nit: it’s not really clear to me the naming of “meta-learner”, what is the meta train/test tasks here? I think something like learned distance metric would be clearer.
>
> **Response:**
> Sorry for the confusion. We now clarify that this draft is not under meta learning settings and there are no such meta train/test tasks. It's true that $f_r$ and $f_d$ learns to measure distances between representations. But the more important roles of $f_r$ and $f_d$ are that they guide the learning of state representations $\phi_r(s)$ and $\phi_d(s)$. As we discussed in Section 4.1, $f_r$ and $f_d$ learn to evaluate the distance between representations in a more flexible space instead of the original Euclidean space,
>
> **Reference**
> - Zhang et al., 2020. Amy Zhang, Rowan McAllister, Roberto Calandra, Yarin Gal, and Sergey Levine. Learning invariant representations for reinforcement learning without reconstruction.
> - Agarwal et al., 2021. Rishabh Agarwal, Marlos C. Machado, Pablo Samuel Castro, and Marc G Bellemare. Contrastivebehavioral similarity embeddings for generalization in reinforcement learning.
>
> Thanks again for your helpful suggestions!

---

### Official Review · Reviewer_Np3q · 2021-11-02

**Correctness:** 2
**Technical Novelty And Significance:** 2
**Empirical Novelty And Significance:** 3
**Recommendation:** 5
**Confidence:** 3

**Main Review:**

*Strengths*

- The paper studies an important problem in learning visual representations for RL robust to distractions.
- The experimental results are pretty strong, outperforming the relevant baselines (DBC, DrQ) and including ablations which show all components of the proposed approach are useful.
- Some components of the method, like combining bisimulation with data augmentation are well motivated.

*Weaknesses*

Despite the ablation which shows that each component of the method is important to performance, I still have some concerns/questions on the correctness of the proposed method.

First, the idea of learning 2 separate state embeddings, one for reward and one for dynamics. The ablation suggests this is very important for good performance, however it seems to produce a metric that is no longer reflective of bisimulation. The bisim metric should capture states which are functionally similar (hence similar in both immediate reward and future state distribution). But by splitting the two up into separate embeddings, and training one only with the reward and one only with dynamics, it seems this functional similarity is no longer captured? The reward representation will only capture immediate reward, while its not clear what the dynamics representation will capture since its only objective is future state similarity in the same dynamics representation space. Is it not possible that the dynamics model would collapse to simple predicting 0 everywhere to minimize its loss?

Second, the intuition given for using a learned distance instead of L1 or L2 is unclear. While it may be easier to optimize, the point of using L1 or L2 is to explicitly structure the embedding space to make RL easier. When instead using an MLP its not clear how the embedding space itself will be shaped, or why it would be shaped in a way that is good for RL. Also, this is an important part of the method and should be clearly described in the main text, instead of in the appendix. Also, the framing of this learned distance as a "meta-learner" seems incorrect. From what I can understand it is simply using an MLP taking in the two embeddings and predicting distance, but not doing any meta learning. The confusing description + the fact that all implementation details about this are only in the appendix makes this component of the method difficult to understand.

I hope the authors can clarify the above. Besides these aspects of the method which seem incorrect, the results appear strong, and if the authors can address the above concerns I'd be open to raising my score.


**Summary Of The Paper:**

The paper studies the problem of learning invariant visual representations for reinforcement learning. Specifically, it builds on prior work Deep Bisimulation for Control (DBC) (Zhang et al), which learns an image representation predictive of rewards and dynamics and uses it with SAC. This paper makes a few modifications on top of DBC, specifically (1) instead of a single state embedding, it learns two separate ones (one for rewards and one for dynamics), (2) instead of using L1 distance in the embedding space to measure distance it instead uses a learned MLP which predicts distance, (3) using gradients from SAC it learns to dynamically adjust the balance between the dynamics weight and reward weight, and (4) adds image augmentation to the training. Experiments indicate on Distracting Control that on 2/4 tasks it matches DrQ and DBC and on the other 2 tasks outperforms them. Ablations suggest that all components are important, particularly (1) and (2).

**Summary Of The Review:**

The paper studies an important problem and has some solid results. However in their current form the major components of the method are not clearly motivated and there are some questions I have about the correctness/justification of the approach.

---

> ### Author Response · Authors · 2021-11-20
> **Response to Reviewer Np3q**
>
> Thanks for your valuable comments and helpful suggestion!
>
> **Review3.1**
> First, the idea of learning 2 separate state embeddings, one for reward and one for dynamics. The ablation suggests this is very important for good performance, however it seems to produce a metric that is no longer reflective of bisimulation. The bisim metric should capture states which are functionally similar (hence similar in both immediate reward and future state distribution). But by splitting the two up into separate embeddings, and training one only with the reward and one only with dynamics, it seems this functional similarity is no longer captured?
>
> **Response:**
> AMBS provides a relaxation form of bisimulation metric. If we sum up two learned metrics, $f_r$ and $f_d$, with weight $c$, the distance between two representations of states can recover bisimulation metric in DBC. Previous methods capture the functional similarity in one embedding, but AMBS separate them into two. If we concatenate two embeddings, it can play the same role as single embedding does. Concatenation is also how we input state representation to SAC policy.
>
> **Review3.2**
> The reward representation will only capture immediate reward, while its not clear what the dynamics representation will capture since its only objective is future state similarity in the same dynamics representation space. Is it not possible that the dynamics model would collapse to simple predicting 0 everywhere to minimize its loss?
>
> **Response:**
> All 0s in dynamics representation is a saddle point of our objective. However, in practice, $\phi_d(s)$ will not converge to all 0s, because neural network $\phi_d$, $f_d$ and $\hat{P}$ are randomly initialized. After initialization, $\phi_d(s)$ is not all 0s (because $\phi_d$ is randomly initialized) and the gradients of dimensions of $\phi_d(s)$ are various ($f_d$ is randomly initialized). To verify $\phi_d(s)$ is not zeros, we record the values of $\phi_d(s)$ during training as shown in Figure 13 in Appendix C.2. The recorded values are$\frac{1}{n_r}||\phi_r(s)||_1$ for reward representation where $n_r$ is the number of dimensions of $\phi_r(s)$, and $\frac{1}{n_d}||\phi_d(s)||_1$ for dynamic representation where $n_r$ is the number of dimensions of $\phi_d(s)$. Although the L1 norm of $\phi_d(s)$ decreases during training, it converges around 0.4-0.5. The L1 norm of $\phi_r(s)$ also decreases in the training procedure. It can verify that dynamic representation $\phi_d(s)$ will not converge to all zeros.
>
> **Review3.3**
> Second, the intuition given for using a learned distance instead of L1 or L2 is unclear. While it may be easier to optimize, the point of using L1 or L2 is to explicitly structure the embedding space to make RL easier. When instead using an MLP its not clear how the embedding space itself will be shaped, or why it would be shaped in a way that is good for RL. Also, this is an important part of the method and should be clearly described in the main text, instead of in the appendix.
>
> **Response:**
> Firstly, Figure 2 demonstrates that AMBS is easier to optimize than L1/L2, which means it is faster for AMBS to learn state representation that encodes similarities. L1/L2 distances lead to larger regression losses in learning representation, resulting in larger gradients with respect to the encoders. While the SAC Q-function objectives remain the same, gradients of L1/L2 may dominate the update of encoders and lead to poor performance in RL.
> Secondly, L1/L2 makes the embedding space structural and more interpretable, but it does not always make RL easier. While L1/L2 is more difficult to optimize (compared to AMBS), they may abandon some features in order to minimize the regression loss even though such features might be important in Q-value prediction.
>
>
> **Review3.4**
> Also, the framing of this learned distance as a "meta-learner" seems incorrect. From what I can understand it is simply using an MLP taking in the two embeddings and predicting distance, but not doing any meta learning.
>
> **Response:**
> Sorry for the confusion. We clarify that our draft is not under meta learning settings. We name $f_r$ and $f_d$ as "meta-learners" because they guides the learning of state representations $\phi_r$ and $\phi_d$. meta-learners $f_r$ and $f_d$ learn to predict the distance and, at the same time, provide gradients to update $\phi_r$ and $\phi_d$, respectively.
>
> **Review3.5**
> The confusing description + the fact that all implementation details about this are only in the appendix makes this component of the method difficult to understand.
>
> **Response:**
> Sorry for the confusion. Due to the limited pages in main text, we didn't state all implementation details in main text. We revise the draft to create a new subsetion 4.4 in the new version, where we describe some important implementation details about state representations $\phi_{ * }$ and meta-learners $f_{ * }$.
> Thanks again for your valuable comments!

---

> > ### Comment · Reviewer_Np3q · 2021-11-30
> > **Reply to authors**
> >
> > Thanks to the authors for their response.
> >
> > Re Response 3.1:
> > I'm not entirely convinced that AMBS "provides a relaxation form of bisimulation metric". My understanding is that an important component of a bisimulation metric is a *single embedding* trained to predictive of both next state distribution and reward. Since it is a single embedding that needs to predict both, it captures everything needed to estimate future rewards, so it should be sufficient for RL.
> > But in AMBS, the dynamics embedding and reward embedding are separate, and do not influence each other in training. i.e. the reward loss does not impact the network which predicts the dynamics embedding. So even if they are concatenated during RL training, theres no guarantee that it captures everything needed to predict future rewards.
> >
> > Re Response 3.2:
> > Similarly, the dynamics embedding predicting all 0s may be a saddle point for the full loss in equation 9, but it minimizes equation 7 correct? And from what I can tell the loss in equation 7 is the only one which influences the dynamics encoder phi_d. So it's still not entirely clear to me why the dynamics embedding should capture anything meaningful.
> >
> > My main issue in both of the last two points is that phi_d does not need to capture anything about reward, and phi_r does not need to capture anything about dynamics correct? If so then I don't see how this captures a bisimulation metric, even if the two are concatenated together when used for policy learning.
> >
> > Re Response 3.3: Thanks this makes sense.
> >
> > Re Response 3.4: Thanks - I still think readers are likely to be confused by the name and recommend the authors rename f_r and f_d.

---

> > > ### Author Response · Authors · 2021-11-30
> > > **Re: Reply to authors**
> > >
> > > - We discuss about the reason that we utilize two embeddings.
> > > Bisimulation metric uses hyper-parameters for linearly combining the reward and dynamic components. For example, DBC and Castro (2020) use 1 and $\gamma$ as the weights, and Ferns et al., (2004) uses $c_R$ and $c_T$. All of them manually assign the values for the linear combination. However we argue that such combination weights should vary in different environments. Meanwhile, Kemertas et al. (2021) provides a value bond for bisimulation metric that no more requires the restriction between combination weights and $\gamma$. Thus we aim to propose learning-based combination weights. In order to learn the combination weight $c$, we learn two embeddings ($\phi_r(s)$ and $\phi_d(s)$) then concatenate $\phi_r(s)$ and $\phi_d(s)$ weighted by $c$ and fed the concatenation into Q-function. In this way, $c$ can be optimized under Q-function loss. Otherwise, with single embedding (as how DBC does) there is no elegant ways to learn such $c$.
> > >
> > > - It is not true that $\phi_d$ captures nothing about rewards and $\phi_r$ captures nothing about dynamics. Embeddings $\phi_r(s)$ and $\phi_d(s)$ are concatenated and fed into Q-function. They are also optimized in Q-function loss (Equation (8)). Because Q-value is the expectation of future rewards, $\phi_r(s)$ and $\phi_d(s)$ somehow learn both rewards and future dynamics. Equation (6) encourages $\phi_r$ to capture more about the immediate rewards and Equation (7) encourages $\phi_d$ to capture more about the dynamics. Besides, $\phi_r(s)$ and $\phi_d(s)$ share convolutional layers in our implementation, as mentioned in Section 4.4 and Appendix B.2. The update of $\phi_r(s)$ and $\phi_d(s)$ can influence each other.
> > >
> > >
> > > - "Similarly, the dynamics embedding predicting all 0s may be a saddle point for the full loss in equation 9, but it minimizes equation 7 correct?"
> > > It minimizes Equation (7) but $\phi_d(s)$ is unlikely to be zeros because $\phi_d(s)$ is also optimized in the Q-function objective (Equation (8)). All 0s for $\phi_d(s)$ does not minimize the overall objective that combines Equation (7) and (8).
> > >
> > > Reference
> > > - Castro (2020). Pablo Samuel Castro. Scalable methods for computing state similarity in deterministic markov decision processes.
> > > - Ferns et al., (2004). Norm Ferns, Prakash Panangaden, and Doina Precup. Metrics for finite markov decision processes.
> > > - Kemertas et al. (2021). Mete Kemertas and Tristan Aumentado-Armstrong. Towards robust bisimulation metric learning.

---

> ### Author Response · Authors · 2021-11-25
> **Thanks for your comments**
>
> Hi Reviewer Np3q,
>
> Thanks for your valuable comments. We have written a detailed response according to your review. We wonder whether it addresses your concerns about our draft. Thanks!

---

### Official Review · Reviewer_rJ9b · 2021-11-02

**Correctness:** 3
**Technical Novelty And Significance:** 3
**Empirical Novelty And Significance:** 2
**Recommendation:** 6
**Confidence:** 5

**Main Review:**

While an interesting extension of DBC from Zhang et al. (2021), I have 3 main concerns with this paper:
1. **Theoretical issues:** My main concern with this paper is in terms of the theoretical justification for learning $c$.
    * The $c$ term in equation (3) (which originally came from Ferns et al., [2004]) is _directly_ tied to $\gamma$. Indeed, this is how the authors of that paper were able to prove the value-function bounds. Castro [2020] only uses the second $c$ term (and sets it equal to $\gamma$), which still yields the value-function bounds. By making the $c$ a learnable parameter, it seems like the connections to the theoretical properties of bisimulation are completely lost.
    * Furthermore, by having a varying $c$ term, it is not at all clear that the underlying metrics even converge to a fixed point!
    * **Post-rebuttal note:** The authors have mostly addressed this in their rebuttal. There are some minor issues with their proofs and I've provided some suggestions.
1.  **Implementation design choices:**
    * Why is $f_r$ even necessary? Rewards are typically fully observed so it's not clear why this needs to be learned at all.
    * The use of $V_*^{(1)}$ in the last term of equation (1) is a strange design decision, and I'm not sure I follow the justification. What is meant by "make a consensus prediction"?
    * **Post-rebuttal note:** The authors have addressed these concerns in their rebuttal.
1. **Statistical significance of results:*** 3 runs is on the lower-side of what should be used for these types of experiments. Further, the authors do not specify what the shaded areas represent in their figures.
    * **Post-rebuttal note:** The authors have promised to run more seeds. I have provided some suggestions in my high-level comment.


---

Some questions/comments for the authors:
1. In the third paragraph of page 2, the authors say "their behavioral distance to the other state representations". What other state representations are being referred to? It is not clear.
1. In the third paragraph of page 2 the authors say "more side information can be preserved". What does "side information" mean, and how is it being preserved?
1. In the third paragraph of page 2 it says "observe that a smaller loss can be obtained". What loss? A smaller loss with respect to what?
1. In the third paragraph of page 2 it says "the approximation precision issue", which issue are you referring to, specifically?
1. Can you clarify what you mean by "least fixed point" below equation 3?
1. In figure 1, are all the $c$s in the figure (there are three of them) all the same?
1. Below Figure 1 the authors say "where $\phi$ is the state encoder", but no state encoder has been introduced.
1. In Section 4.1, do $\phi_r$ and $\phi_d$ share parameters? Figure 1 suggests they do.
1. In page 5 it says "lead to large regression losses". What regression loss is this, specifically?
1. In page 5 it says "which destabilze the representation learning". In what way is it destabilized?
1. In page 5 it says "which however may loss[sic] part of useful information". What useful information is being referred to, specifically?
1. In the second paragraph of page 5 it says "able to preserve more task-relevant information". What do the authors mean by this, specifically? What task-relevant information?
1. The sentence immediately above equation (11) ends with "comprises of encoder is". What is meant by this?
1. In the line below equation (11) what is meant by "a convention form of Q-function"?
1. Below equation (11), if Q depends on $c$, shouldn't $c$ also be a parameter of the Q function?
1. In the learning curves, what do the shaded areas represent?
1. In section 5.2 it says "**AMBS + Drq** is DBC with data augmentation", do the authors mean "AMBS with data augmentation"?
1. In section 5.2 the authors tried fixing $c$ to $0.5$, but a more natural choice would have been $\gamma$, given the main point made above. Further, neither DBC nor the original $\pi$-bisimulation from Castro (2020) use the $(1-c)$ term in front of the reward differences. Fixing $c=\gamma$ and removing the $(1-c)$ term would have yielded a more direct comparison to DBC.

---

Some minor comments:
1.  In the first paragraph of the introduction, consider replacing "based on some RL algorithms" with "based on various RL algorithms"
1.  In the first line of page 2, should be "of **a** Markov Decision Process"
1.  A paper from early in the year is quite relevant and should be included in the related work section for representation learning:
    * [Agarwal et al., Contrastive Behavioral Similarity Embeddings for Generalization in Reinforcement Learning; ICLR 2021](https://arxiv.org/abs/2101.05265).
1.  In the second paragraph of page 2, consider including two recent papers that have been accepted to NeurIPS:
     *  [Castro et al., MICo: Improved representations via sampling-based state similarity for Markov decision processes](https://arxiv.org/abs/2106.08229)
     *  [Kemertas and Aumentado-Armstrong, Towards Robust Bisimulation Metric Learning](https://arxiv.org/abs/2110.14096)
1.  In the second paragraph in page 2, should be "and potentially ***lose*** parts of ***the*** state features"
1.  In the third paragraph in page 2 they say "pair of meta-learners that learn similarities". Please specify which similarities are being referred to.
1.  In the third paragraph in page 2 it should say "hand-craft***ed*** form"
1.  Right above section 3, use `\citet` for "Pitis et al".
1. In section 3, specify the range of $\gamma$. e.g. $\gamma\in [0, 1)$
1. Right below equation 2, should say "where $d$ quantif***ies*** the behavioral..."
1.  In the same paragraph, better to say "defines a metric ***with respect*** to a policy $\pi$.
1. In the caption of Figure 1, should say "which ***is*** jointly learned"
1. Below Figure 1 caption, $\gamma$ is referred to "the hyper-parameter", but I think the authors mean discount factor.
1. Below Figure 1 caption, should say "combined with ***the*** reinforcement learning..."
1. In the first paragraph of section 4, remove the "the" before Figure 1 (i.e. "is demonstrated in Figure 1")
1. In the first paragraph of section 4, should say "two learned similarit***ies*** in a specific..."
1. In section 4.1 the network architecture is mentioned but has not been introduced. I'd suggest referencing the appendix where it is introduced.
1. In the second paragraph of page 5 it should say "and therefore ***the*** meta-learner..."
1. In the second paragraph of page 5 it should say "the process of udpating ***the*** state encoder"
1. In the second paragraph of page 5 it should say "Besides, $f_*$ is ***a*** non-linear transformation"
1. In equation (5) specify that you are using the closed form of the $W_2$ metric (as in DBC). This is mentioned in the appendix, but should be clarified in the main paper as well.
1. Above equation (6) you mention the "learned parameteric dynamics model". Please add a reference to the appendix where it is defined.
1. Above equation (6) should say "two transitions sampled from ***the*** replay buffer"
1. In the first paragraph of 4.2 you should specify that $c\in (0, 1)$ is enforced by a softmax. This is specified in the appendix but should be clarified in the main paper as well.
1. In the paragraph above equation (11) should say "we aim to make $f_r$ and $f_d$ ***symmetric***"
1. In the line above 5.1, should say "which is a common***ly*** use***d*** off-policy..."
1. In the first sentence in 5.1, should say "is a***n*** environment ***that*** provides high dimensional pixel observation***s*** for RL tasks"
1. In the first sentence of 5.3 it should say "but have differen***t*** reward functions"
1. In section 5.5 should say "learning of RL agent***s*** may suffer from..."
1. In section 5.5 should say "DBC to create a***n*** autonomous driving..."

**Summary Of The Paper:**

This paper leverages some recent work on bisimulation metrics to develop a pair of meta-learners to capture the two parts of a bisimulation metric: reward similarities and dynamics similarities. The authors evaluate their method on the environments used by the recent DBC (Zhang et al., 2021) paper, and compare against this and other related methods.

**Summary Of The Review:**

See main concerns above.

---

> ### Author Response · Authors · 2021-11-20
> **Response to Reviewer rJ9b (1/3)**
>
> Thanks for your valuable comments and helpful suggestion.
>
> **Review2.1**
> Theoretical issues: My main concern with this paper is in terms of the theoretical justification for learning $c$.
> The $c$ term in equation (3) (which originally came from Ferns et al., [2004]) is directly tied to $\gamma$. Indeed, this is how the authors of that paper were able to prove the value-function bounds. Castro [2020] only uses the second $c$ term (and sets it equal to $\gamma$), which still yields the value-function bounds. By making the $c$ a learnable parameter, it seems like the connections to the theoretical properties of bisimulation are completely lost.
>
> **Response:**
> Ferns et al., [2004] proved value-function bounds given $\gamma \leq c$ and Castro [2020] proved the value-function bounds given $\gamma = c$. In our method, we don't restrict $c$ with the discounted factor $\gamma$, but there is still a value-function bound.
> Let $d$ denotes a bisimulation metric with coefficients $(1-c)$ and $c$. It satisfies a value-function bound that is $(1-c)|V^{ * }(s)-V^{ * }(s^{\prime})| \leq d(s, s^{\prime}) + \frac{2(1-c)(\gamma-\min(c, \gamma))}{(1-\gamma)(1-c)}$. It can be proved by following Theorem 1 in Kemertas et al. [2021].
> We add the proof of value-function bound in Appendix E.
>
> **Review2.2:**
> Furthermore, by having a varying $c$ term, it is not at all clear that the underlying metrics even converge to a fixed point!
>
> **Response:**
> Based on Theorem 4.5 in Ferns et al., [2004], it doesn't require $c$ to be $\gamma$ for the proof of the existence of fixed-point. In our method, $c$ is not updated by the gradient from objective (10), which means that $c$ is still a constant for objective (10). Although $c$ is varying during the training, for each value of $c$ it still satisfies Theorem 4.5 in Ferns et al., [2004].
>
> **Review2.3**
> Why is $f_r$ even necessary? Rewards are typically fully observed so it's not clear why this needs to be learned at all.
>
> **Response:**
> $f_r$ learns to measure the distance between two representations $\phi_r(s_i)$ and $\phi_r(s_j)$ by regressing to reward difference $|r_i-r_j|$. The final outcome of our method is to learn generalized state representations for RL. The role of $f_r$ is to guide the learning of representation $\phi_r(s)$. Although reward signals ($r_i$ and $r_j$) are fully observed, we still need to learn an encoder to encode a state into a vector ($\phi_r(s)$) regarding reward signal.
>
> **Review2.4**
> The use of $V_{*}^{(1)}$ in the last term of equation (1) is a strange design decision, and I'm not sure I follow the justification. What is meant by "make a consensus prediction"?
>
> **Response:**
> I suppose you are talking about equation (10). Look into equation (10), there are two terms regarding to your question, the second term $c \bigg(f_d \left(\phi_d(\mathbf{s}_i^{(1)}), \phi_d(\mathbf{s}_j^{(1)})\right) - W_2\left(\hat{\mathcal{P}}(\cdot|\phi_d(\mathbf{s}_i^{(1)}), \mathbf{a}_i), \hat{\mathcal{P}}(\cdot|\phi_d(\mathbf{s}_j^{(1)}), \mathbf{a}_j)\right)\bigg) ^ 2$ and the last term $c \bigg(f_d\left(\phi_d(\mathbf{s}_j^{(2)}), \phi_d(\mathbf{s}_i^{(2)})\right) - W_2\left(\hat{\mathcal{P}}(\cdot|\phi_d(\mathbf{s}_i^{(1)}), \mathbf{a}_i), \hat{\mathcal{P}}(\cdot|\phi_d(\mathbf{s}_j^{(1)}), \mathbf{a}_j)\right)\bigg) ^ 2$.
>
> Both of them are regression losses and they have the same regression target
> $ W_2\left(\hat{\mathcal{P}}(\cdot|\phi_d(\mathbf{s}_i^{(1)}), \mathbf{a}_i), \hat{\mathcal{P}}(\cdot|\phi_d(\mathbf{s}_j^{(1)}), \mathbf{a}_j)\right) $,
>
> where $v^{(1)}_{ * }$ is used. Because they have the same regression target, $f_d (\phi_d(\mathbf{s}_i^{(1)}), \phi_d(\mathbf{s}_j^{(1)}))$ and $f_d(\phi_d(\mathbf{s}_j^{(2)}), \phi_d(\mathbf{s}_i^{(2)}))$ ,
>
> which have different data augmentation parameter $v_{ * }$, learns to predict the same distances. Therefore we claim that $f_d$ "make a consensus prediction" over different data augmentation parameter $ v_{ * } $.
>
> **Review2.5**
> Statistical significance of results: 3 runs is on the lower-side of what should be used for these types of experiments. Further, the authors do not specify what the shaded areas represent in their figures.
>
> **Response:**
> Due to our machine capacity, only 3 runs are shown in this draft.  We will run more runs for the experiments. The shaded areas are the standard deviation. Sorry for the confusion.
>
> **Review2.6**
> In the third paragraph of page 2, the authors say "their behavioral distance to the other state representations". What other state representations are being referred to? It is not clear.
>
> **Response:**
> Sorry for the confusion. "Other state representations" should be referred to "representations of other states". Here it means the behavioral distance between representations of two states.

---

> > ### Comment · Reviewer_rJ9b · 2021-11-24
> > **Response 1/3**
> >
> > Thank you for clarifying these points.
> >
> > **2.1:** Theorem E.3 mostly addresses my concerns with $c$. However, I think there is a minor bug in the proof. One of the conditions of Theorem E.2 is that $c\geq \gamma$. Thus in your new bound the additive term actually goes away, since $(\gamma - min(c, \gamma)) = (\gamma - \gamma) = 0$.
> >
> > This does satisfy my main concern, although it seems like based on this result you should modify your algorithm so that the values of $c$ are restricted to be in $(\gamma, 1)$ as opposed to $(0, 1)$.
> >
> > **Other points:** Thanks for clarifying. Worth adding the discussions you put here to clarify for other readers!

---

> > > ### Author Response · Authors · 2021-11-25
> > > **Re: Response 1/3**
> > >
> > > Let us clarify the reason that $c$ is in the interval  $(0, 1)$.
> > >
> > > In the proof of Theorem E.3, we define an auxiliary discounting factor $\gamma^{\prime}=c$, which satisfies the condition $c \geq \gamma^{\prime}$ in Theorem E.2. We apply Theorem E.2 at the fourth line of Equation (24), with respect to $ \gamma^{\prime} $ instead of the original discounting factor $\gamma$, to obtain $ d(s_i,s_j) $. This eliminates the restriction between $c$ and $\gamma$. Therefore the value of $c$ is in the interval $(0, 1)$.
> > >
> > > We discuss a bit more about the additive term.
> > >
> > > At the last line of Equation (24), we apply Theorem E.1 to obtain the additive term $\frac{2(1-c)(\gamma-\min(\gamma^{\prime}, \gamma))}{(1-\gamma)(1-\gamma^{\prime})} $ , where $\min(\gamma^{\prime}, \gamma)$ is to satisfy the condition $\gamma^{\prime} \leq \gamma$ in Theorem E.1.
> > > Such additive term can be considered as a correction for the case that $c \in (0, \gamma)$( or $\gamma^{\prime} < \gamma $).
> > > In the other case of $c \in [\gamma, 1) $ (or $\gamma^{\prime} \geq \gamma$), the additive term is zero so that the bound in Theorem E.3 reduces to Theorem E.2.

---

> > > > ### Comment · Reviewer_rJ9b · 2021-11-25
> > > > **Re: Re: Response 1/3**
> > > >
> > > > Ah, I see, thanks for clarifying. Could be useful to be more explicit for the final version to avoid other confused readers like me :).
> > > >
> > > > Minor point, there's a left parenthesis and a right absolute value missing in the fourth line of equation (24).

---

> ### Author Response · Authors · 2021-11-20
> **Response to Reviewer rJ9b (2/3)**
>
>
> **Review2.7**
> In the third paragraph of page 2 the authors say "more side information can be preserved". What does "side information" mean, and how is it being preserved?
>
> **Response:**
> Encoding in L1 space leads to larger loss and results in being more difficult to optimize the state representation (as shown in Figure 2).  In order to minimize the regression loss, the encoder may discard some information not relative to measure distance, although such information might be important to predict Q-value for RL. AMBS is easier to optimize and does not restrict embeddings to L1 space, so it may be likely to preserve such information to make RL agent learning faster.
>
>
>
> **Review2.8**
> In the third paragraph of page 2 it says "observe that a smaller loss can be obtained". What loss? A smaller loss with respect to what?
>
> **Response:**
> The "loss" is the regression loss for the representation. For DBC, the regression loss is the last equation in page 3. For our method, the regression loss is equation (10). The comparison of losses is shown in Figure 2 in Section 4, page 5.
> The regression loss of AMBS is smaller than DBC using L1 distance.
> More comparison of losses in other environments is shown in Appendix C.1. They show the similar tendency between AMBS and DBC.
>
> **Review2.9**
> In the third paragraph of page 2 it says "the approximation precision issue", which issue are you referring to, specifically?
>
> **Response:**
> "The approximation precision issue" is referred to the increasing regression error in DBC when using L1 distance. We illustrate the curves of such regression loss in Figure 2 and Figure 10. Our AMBS utilizing meta-learner can overcome such issue.
>
> **Review2.10**
> Can you clarify what you mean by "least fixed point" below equation 3?
>
> **Response:**
> The "least fixed point" property is claimed by Castro [2020] (Theorem 2 in their paper).
> "Least fixed point" property means that if we iterate the transformation $d_{n+1}=F^{\pi}(d_n)$, it will eventually converge to a pseudometric $d_*$ that $d_*=F^{\pi}(d_*)$.
>
> **Review2.11**
> In figure 1, are all the $c$s in the figure (there are three of them) all the same?
>
> **Response:**
> Yes, all $c$s are referred to the same parameters.
>
> **Review2.12**
> Below Figure 1 the authors say "where $\phi$ is the state encoder", but no state encoder has been introduced.
>
> **Response:**
> A single encoder $\phi$ is used in DBC paper(Zhang et al., 2020) and they use CNN to implement $\phi$. Our AMBS uses two state encoders $\phi_r$ and $\phi_d$ which are introduced in Section 4.1. The detail of implementation is described in Appendix B.2.
>
> **Review2.13**
> In Section 4.1, do $\phi_r$ and $\phi_d$ share parameters? Figure 1 suggests they do.
>
> **Response:**
> Yes, they shared the convolutional layers. We mentioned it in Appendix B.2.
>
> **Review2.14**
> In page 5 it says "lead to large regression losses". What regression loss is this, specifically?
>
> **Response:**
> For AMBS, regression losses are referred to Equation (6) and (7). Regression losses for L1 or L2 are referred to the DBC objective described as the last equation in Section 3.
>
>
> **Review2.15**
> In page 5 it says "which destabilze the representation learning". In what way is it destabilized?
>
> **Response:**
> While L1/L2 distance have larger regression losses, the gradients with respect to encoder is also larger. Obviously the state encoders are difficult to optimize and the state representations are also difficult to learn.
>
>
> **Review2.16**
> In page 5 it says "which however may loss[sic] part of useful information". What useful information is being referred to, specifically?
> In the second paragraph of page 5 it says "able to preserve more task-relevant information". What do the authors mean by this, specifically? What task-relevant information?
>
> **Response:**
> In order to minimize the regression loss for L1/L2, the encoder may discard some information not relative to measure distance, although such information might be important to predict Q-value for RL. AMBS is easier to optimize and does not restrict embeddings to L1 space, it may be likely to preserve such information to make RL agent learning faster.
>
> **Review2.17**
> The sentence immediately above equation (11) ends with "comprises of encoder is". What is meant by this?
>
> **Response:**
> The original SAC actor objective takes input of state. But is our method, the actor in equation (11) takes input of representation $\hat{\phi}(s)$.

---

> > ### Comment · Reviewer_rJ9b · 2021-11-24
> > **Response 2/3**
> >
> > **2.7:** Thank you for clarifying. It would be better to make the language a bit more precise to avoid these confusions, as on first reading it's not clear what "side information" means.
> >
> > **2.8, 2.9, 2.14:** Thanks for clarifying, would be nice if you could clarify in paper.
> >
> > **2.12:** Probably worth adding a pointer to that when it is introduced.
> >
> > **2.13:** Thanks, probably best to specify this in paper.
> >
> > **2.15:** Ok, would be good to clarify this in paper.
> >
> > **2.16:** This seems a little tenuous. In particular, it's not clear what "information" means in this case (beyond a high-level intuition). Given that this statement is not very precise, I'd suggest making this less of a claim and more of a high-level hypothesis, and perhaps best in the discussion.
> >
> > **2.17:** Ok. The language needs some correcting then, as it's not grammatically correct. I'm still not completely sure what you're trying to say, but perhaps something like "The loss of the actor of SAC, which is based on the output of the encoder, is"
> >
> > **Other points:** Thanks for clarifying!

---

> > > ### Author Response · Authors · 2021-11-25
> > > **Re: Response 2/3**
> > >
> > > Thanks for your helpful suggestion. We will revise our paper in the final version accordingly.
> > >
> > > **2.7** **2.8** **2.9** **2.14** **2.15**:
> > > We will clarify in the final version.
> > >
> > > **2.12**  **2.13**:
> > > We will describe them more in the paper.
> > >
> > > **2.16**: We will explain and discuss it in the final version.
> > >
> > > **2.17**: Yes, that is what we mean. We will revise this sentence.

---

> ### Author Response · Authors · 2021-11-20
> **Response to Reviewer rJ9b (3/3)**
>
> **Review2.18**
> In the line below equation (11) what is meant by "a convention form of Q-function"?
>
> **Response:**
> We use DrQ data augmented SAC in our method, therefore equation (11) should includes augmented state $\mathbf{s}^(1)$ and weight $c$ (as Equation (8)). Due to the limited pages of main text, we use $Q_{\theta}(\phi(\mathbf{s}),\mathbf{a})$ for short. The full objective of (11) is described Equation (16) in Appendix A.
>
> **Review2.19**
> Below equation (11), if Q depends on $c$, shouldn't $c$ also be a parameter of the Q function?
>
> **Response:**
> We consider $c$ as input of $Q$ as shown in Equation (8). Equation (8) also shows that $Q$ takes input of $c$ and $c$ is a parameter in objective $J_Q(\theta, c)$. Notes that the encoders $\phi_r$ and $\phi_d$ are being updated in Equation (8) but $\phi_r(s)$ and $\phi_d(s)$ are input of $Q$. For consistency, we also consider $c$ as input of $Q$. We also revise Equation (8) to clarify  $\phi_r$ and $\phi_d$ are received gradients from Equation (8).
>
> **Review2.20**
> In the learning curves, what do the shaded areas represent?
>
> **Response:**
> The shaded areas are the standard deviation over 3 runs.
>
>
> **Review2.21**
> In section 5.2 it says "**AMBS + Drq** is DBC with data augmentation", do the authors mean "AMBS with data augmentation"?
>
> **Response:**
> Sorry for the typo. It should be **DBC + Drq**, to be consistent to the legends in Figure 6. We revise this term in the new version of draft.
>
> **Review2.22**
> In section 5.2 the authors tried fixing $c$ to $0.5$, but a more natural choice would have been $\gamma$, given the main point made above. Further, neither DBC nor the original $\pi$-bisimulation from Castro (2020) use the $(1-c)$ term in front of the reward differences. Fixing $c=\gamma$ and removing the $(1-c)$ term would have yielded a more direct comparison to DBC.
>
> **Response:**
> We add a new setting named **AMBS w/ (1,$\gamma$)** which use $1$ for reward difference and $\gamma$ for dynamic difference in ablation study in Section 5.2. Figure 6 shows that **AMBS w/ (1,$\gamma$)** uderperforms **AMBS** in the cheetah-run and walker-walk. **AMBS** learns faster than **AMBS w/ (1,$\gamma$)** in both environments and converge to better scores in cheetah-run.
>
> **Review2.23**
> Some minor comments:...
>
> **Response:**
> Thanks for your helpful suggestions. We revise our draft accordingly to address these typos and grammar issues. We also carefully proofread the draft again and revise typos and grammar errors.
>
> **Review2.24**
> A paper from early in the year is quite relevant and should be included in the related work section for representation learning:
> 1) Agarwal et al., Contrastive Behavioral Similarity Embeddings for Generalization in Reinforcement Learning; ICLR 2021.
>
> In the second paragraph of page 2, consider including two recent papers that have been accepted to NeurIPS:
> 1) Castro et al., MICo: Improved representations via sampling-based state similarity for Markov decision processes
> 1) Kemertas and Aumentado-Armstrong, Towards Robust Bisimulation Metric Learning
>
> **Response:**
> We have add all of three papers to relative works section.
>
> Thanks again for your helpful suggestions!
>
> **Reference**
>
> - Ferns et al., [2004]. Norm Ferns, Prakash Panangaden, and Doina Precup. Metrics for finite markov decision processes.
> - Castro [2020]. Pablo Samuel Castro. Scalable methods for computing state similarity in deterministic markov decision processes.
> - Kemertas et al. [2021]. Mete Kemertas and Tristan Aumentado-Armstrong.  Towards robust bisimulation metric learning.

---

> > ### Comment · Reviewer_rJ9b · 2021-11-24
> > **Response 3/3**
> >
> > **2.18:** Ok, I would suggest rewording it, perhaps something like: "and $Q_{\theta}(\phi(s), a)$ is used as a shorthand for a Q-function
> > that takes $c$-weighted state representations as input (see Appendix A for the full objective)."
> >
> > **Other coments:** Thanks for clarifying!

---

> > > ### Author Response · Authors · 2021-11-25
> > > **Re: Response 3/3**
> > >
> > > Thanks for your helpful suggestion. We will revise our paper accordingly in the final version.

---

> ### Comment · Reviewer_rJ9b · 2021-11-24
> **Revising score**
>
> The authors have mostly addressed my first two main concerns (theoretical and implementation issues), so I will revise my score accordingly.
> The third concern (statistical significance) has not yet been addressed, but the authors have promised to do so for the final version, I would suggest _at least_ 5 runs are needed for each algorithm/env combination. I can appreciate that access to compute might be an issue, and if so, I would suggest prioritizing which experiments get the larger numbers of seeds (e.g. Figs 4 and 5 are probably higher priority than Fig 6).

---

> > ### Author Response · Authors · 2021-11-25
> > **Re: Revising score**
> >
> > Thanks for your constructive comments. We will perform more runs for each algorithm in experiments. We will show the curves in the final version.

---

### Official Review · Reviewer_vhfp · 2021-11-03

**Correctness:** 3
**Technical Novelty And Significance:** 3
**Empirical Novelty And Significance:** 3
**Recommendation:** 6
**Confidence:** 4

**Main Review:**

## Strengths
- Overall, the approach proposed in this work seems technically sound and well-motivated. Learning a state representation based on a bisimulation metric seems quite reasonable, and doing so appears to yield strong empirical results.
- The experimental evaluation is quite thorough and illustrates several interesting phenomena. I appreciate that this work both considers environments where most of the state observation includes only distracting information in the DM Control tasks, as well as environments where a large part of the state observation is potentially useful, as in the CARLA tasks. The ablation studies and results on transferring to different reward functions are also interesting, and present fairly promising results.
- The paper is generally fairly clear.

## Weaknesses
- There are no critical or significant weaknesses that I am aware of, though potentially slightly incremental on top of DBC.
- There are numerous minor typos and small errors throughout the paper. For example, the domain and range of the function $F^\pi$ in Equation (3) appear to be some undefined "met" space. At the end of the preliminaries, "partial observations" is used, where I believe "partial observability" is meant. The equations (4) and (5) are defined as "least squares error," where I believe "mean squared error" is meant, or this could refer to a form of least squares regression. I believe that these are generally easy to address, but it would be useful to clean this up.

## Additional Questions / Comments
- In Equations (6) and (7), shouldn't this be restricted to samples where $a_i = a_j$? If the two actions don't match, I don't think you can necessarily expect the rewards and dynamics to be similar, even in similar states, e.g., even in exactly the same state. This seems potentially problematic to me.
- Why does the actor condition only on the CNN layers of $\phi$ in Equation (11) and not on the full learned state representation like the critic?
- Do the representations $\phi_r$ and $\phi_s$ also receive gradients through updating the critic in $J_Q$? Or are they learned exclusively via the $\ell(\Theta)$ loss?

-------

## Post-rebuttal comments

I appreciate the rebuttal, which has alleviated the concerns I've raised. I still find the proposed approach technically sound and well-motivated, and find that this work provides a reasonable empirical contribution. Several of the other reviewers also raised concerns about whether the approach is empirically justified. I find these concerns to be reasonable, although they are partially alleviated via the rebuttal and I believe they are outweighed by the empirical contribution of the work. Therefore, I continue to recommend acceptance, though I do not feel strongly enough to raise my score to the next possible option, which is an 8.

**Summary Of The Paper:**

This work considers the problem of learning representations of high-dimensional pixel observations for RL. High-dimensional pixel observations often include many task-irrelevant details and ideally, an effective representation of such observations should only encode the task-relevant details. To implement this intuition, this work proposes to learn a two-part representation based off of a bisimulation metric. Two states that have similar first parts of the representation should exhibit similar reward structure, and two states that have similar second parts of the representation should exhibit similar dynamics structure. This work evaluates this learned representation on several continuous control tasks with distracting backgrounds and on CARLA, and shows that the proposed learned representation improves over baselines.

**Summary Of The Review:**

Overall, this work offers a principled and empirically strong approach. There are some weaknesses, but they seem relatively easy to address, and the contributions of the work outweigh the cons. There were also some points of confusion for me in the paper that seem potentially serious, which I included above. I initially recommend a score of 6, but am willing to raise my score if the points of confusion are clarified.

---

> ### Author Response · Authors · 2021-11-20
> **Response to Reviewer vhfp**
>
> Thanks for your constructive comments and appreciation of our work.
>
> **Review1.1**
> There are no critical or significant weaknesses that I am aware of, though potentially slightly incremental on top of DBC.
>
> **Response:**
> In this manuscript, we propose a framework that learns behavioral similarities and learn generalizable state representations for deep RL. We observed that DBC suffers from the issues of approximating the bisimulation metric by L1 distance, then we propose AMBS using two neural network-based meta-learners, which are able to learn behavioral similarities faster and achieve lower approximation errors. Besides, we use two state encoders that learn two separate state embeddings regarding reward and dynamic, respectively, and design a learnable adaptive strategy to balance them, while DBC learns a single representation for each state.
>
> **Review1.2**
> The domain and range of the function $F^*$ in Equation (3) appear to be some undefined "met" space.
>
> **Response:**
> Sorry for the confusion. The "met" denotes the set of all pseudometrics on state space $\mathcal{S}$ and $F^*$ is a mapping from a metric to another metric. We have revised this part in the new version of manuscript.
>
> **Review1.3**
> At the end of the preliminaries, "partial observations" is used, where I believe "partial observability" is meant. The equations (4) and (5) are defined as "least squares error," where I believe "mean squared error" is meant, or this could refer to a form of least squares regression. I believe that these are generally easy to address, but it would be useful to clean this up.
>
> **Response:**
> Thanks for pointing out these errors in our draft. We have revised to "partial observability" and "mean squared error" according to your advise.
>
> **Review1.4**
> In Equations (6) and (7), shouldn't this be restricted to samples where $a_i=a_j$
> ? If the two actions don't match, I don't think you can necessarily expect the rewards and dynamics to be similar, even in similar states, e.g., even in exactly the same state. This seems potentially problematic to me.
>
> **Response:**
> Actions $a_i$ and $a_j$ don't have to be match. When action space is continuous, it is unlikely to sample two exactly the same actions.
> The action in a transition is sampled from the policy before such transition is added into the replay buffer. If two states are similar, then the action distributions $\pi(a|s)$ would be similar as well.  The actions sampled from similar action distributions are more likely to be close to each other.
> Although we cannot expect that similar states necessarily lead to similar rewards and dynamics for all transitions, we can expect Equations (6) and (7) can still learn metrics from the overall distribution.
>
> **Review1.5**
> Why does the actor condition only on the CNN layers of $\phi$ in Equation (11) and not on the full learned state representation like the critic?
>
> **Response:**
> We have tried sharing the full representation $\phi(s)$ to the actor before. However, it performs much worse than only sharing CNN layers of $\phi$.
>
> **Review1.6**
> Do the representations $\phi_r$ and $\phi_s$ also receive gradients through updating the critic in $J_Q$? Or are they learned exclusively via the $l(\Theta)$ loss?
>
> **Response:**
> Yes, $\phi_r$ and $\phi_s$ receive gradients from critic objective $J_Q$. The encoders $\phi_r$ and $\phi_s$ should also capture to learn Q-values. We revise the objective $J_Q$ (Equation (8)) to clarify it in the new version of our draft.
>
> Thanks again for your helpful suggestions!

---

> > ### Comment · Reviewer_vhfp · 2021-11-26
> > **Reply**
> >
> > Thanks for the helpful response. This alleviates most of my concerns.
> >
> > I still have concerns about Equations (6) and (7). If the actions don't necessarily match, then these loss functions seem to lead to potentially problematic behavior for the representations. For example, if $s_i$ and $s_j$ are actually states with identical dynamics and rewards, but $a_i \neq a_j$ in such a way that $|r_i - r_j|$ is large, this seems to cause the representations $\phi(s_i)$ and $\phi(s_j)$ to be pushed far apart, which seems undesirable. Similarly, this can occur in Equation (7) if the dynamics from $a_i$ and $a_j$ are quite different. Furthermore, I imagine that it's not uncommon that there are several states with similar dynamics and rewards, and their dynamics and rewards are quite different for each action. Can you clarify how this is okay?

---

> > > ### Author Response · Authors · 2021-11-28
> > > **Re: Reply**
> > >
> > > The Equations (6) and (7) are to learn metrics to distinguish similar and dissimilar state pairs: small distances for similar pairs and large distances for dissimilar pairs. If $s_i$ and $s_j$ are states with identical dynamics and rewards, it is true that the learned distance between $s_i$ and $s_j$ is not necessarily zero, but small enough to distinguish with other dissimilar state pairs. As $s_i$ and $s_j$ are semantically identical, the action distribution $\pi(a_i|s_i)$ and $\pi(a_j|s_j)$ should be the same. We assume policy $\pi$ is Gaussian distribution in practice. Then $(a_i - a_j) \sim N(0, \sqrt{2}\sigma)$ (because of Gaussian distribution property) where $\sigma$ is the standard deviation of $\pi(a|s)$. During RL training, the variance of the policy $\sigma^2$  decreases to a small value (but not to zero because of SAC maximizing entropy).
> > > The sampled actions $a_i$ and $a_j$ are likely to be close to each other as $\sigma$ is small, therefore $|r_i - r_j|$ is also likely close to zero.
> > > Although it may sample distinct action pairs ($|a_i - a_j|$ is large) which lead to large $|r_i - r_j|$, the probabilities of such pairs are small. The expectation of reward difference $E_{a_i, a_j \sim \pi(a|s)}[|r_i - r_j|]$ remains close to zero. The training samples for Equation (6) are randomly sampled and finally $f_r$ learns to approximate $E[|r_i - r_j|]$. Therefore $f_r$ does not push representation $\phi_r(s_i)$ and $\phi_r(s_j)$ far apart.

---

> > > > ### Comment · Reviewer_vhfp · 2021-11-28
> > > > **Reply**
> > > >
> > > > Please correct me if I am missing something, but this argument feels somewhat circular.
> > > >
> > > > > As $s_i$ and $s_j$ are semantically identical, the action distribution $\pi(a_i \mid s_i)$ and $\pi(a_j \mid s_j)$ should be the same.$
> > > >
> > > > If I understand correctly, the policy $\pi(a \mid s)$ conditions on the learned representation $\phi(s)$, and therefore the action distribution at two states $s_i$ and $s_j$ is only going to be the same if the learned representations $\phi(s_i)$ and $\phi(s_j)$ is the same. This is unlikely to be the case upon initialization, and when this is not true, Equations (6) and (7) seems to push the representations apart.

---

> > > > > ### Author Response · Authors · 2021-11-29
> > > > > **Re: Reply**
> > > > >
> > > > > The policy $\pi$ conditions on representation $\phi(s)$ but the learning of policy $\pi$ does not fully depend on representation $\phi(s)$. Consider a RL algorithm, e.g., SAC or DrQ, which learns without AMBS nor other bisimulation metric methods. Such RL algorithm can still learn a policy that outputs similar action distributions $\pi(a_i|s_i)$ and  $\pi(a_j|s_j)$ for state $s_i$ and $s_j$ even though it may take more training steps, otherwise $s_i$ and $s_j$ are not similar states.
> > > > > AMBS accelerates such learning procedure of RL policy.
> > > > > Furthermore, the different representations $\phi(s_i)$ and  $\phi(s_j)$ can be considered as data augmentation for policy learning.
> > > > >
> > > > > Because we focus on pixel-input RL tasks in this paper, it is unlikely to sample two states that are exactly identical on pixels. Even though two states are semantically identical, i.e., with identical rewards and dynamics, they got differences on pixels because of RL tasks with background distraction. Therefore $\phi(s_i)$ and $\phi(s_j)$ for semantically identical states $s_i$ and $s_j$ would be close but not the same. Equations (6) and (7) push such representations apart but they don't push such representations far away. The distance between $\phi(s_i)$ and $\phi(s_j)$ would be small enough to distinguish with other dissimilar state pairs.

---

> > > > > > ### Comment · Reviewer_vhfp · 2021-11-30
> > > > > > **Reply**
> > > > > >
> > > > > > Understood. That makes sense. Thanks for the clarification!

---

### Decision · Program_Chairs · 2022-01-20

**Decision:**

Accept (Poster)

**Comment:**

The paper proposes to learn a state-representation using bi-simulation in an RL setting. The approach is thoroughly evaluated on several benchmarks. In its current form the paper is mainly an empirical contribution, with now some theoretical contributions tucked away in the appendices. Nevertheless, an interesting approach with promising results.

The reviewers appreciated the revised paper and the discussion. The replies and discussions successfully addressed all serious concerns of the reviewers. Please also clarify the discussed points in the next iteration of the paper, and run the experiments with more seeds, as promised.